# MEMO1 binds iron and modulates iron homeostasis in cancer cells

Natalia Dolgova[1†], Eva-Maria E Uhlemann[1†], Michal T Boniecki[2†], Frederick S Vizeacoumar[3†], Anjuman Ara[1], Paria Nouri[1], Martina Ralle[4], Marco Tonelli[5], Syed A Abbas[1], Jaala Patry[1], Hussain Elhasasna[3], Andrew Freywald[3], Franco J Vizeacoumar[6,7]*, Oleg Y Dmitriev[1]*

[1]Department of Biochemistry, Microbiology and Immunology, University of Saskatchewan, Saskatoon, Canada; [2]Protein Characterization and Crystallization Facility, University of Saskatchewan, Saskatoon, Canada; [3]Department of Pathology and Laboratory Medicine, University of Saskatchewan, Saskatoon, Canada; [4]Department of Molecular and Medical Genetics, Oregon Health and Sciences University, Portland, United States; [5]National Magnetic Resonance Facility at Madison (NMRFAM), University of Wisconsin, Madison, United States; [6]Cancer Research Department, Saskatchewan Cancer Agency, Saskatoon, Canada; [7]Division of Oncology, University of Saskatchewan, Saskatoon, Canada

*For correspondence:
franco.vizeacoumar@usask.ca (FJV);
Oleg.Dmitriev@usask.ca (OYD)

[†]These authors contributed equally to this work

**Abstract** Mediator of ERBB2-driven cell motility 1 (MEMO1) is an evolutionary conserved protein implicated in many biological processes; however, its primary molecular function remains unknown. Importantly, MEMO1 is overexpressed in many types of cancer and was shown to modulate breast cancer metastasis through altered cell motility. To better understand the function of MEMO1 in cancer cells, we analyzed genetic interactions of MEMO1 using gene essentiality data from 1028 cancer cell lines and found multiple iron-related genes exhibiting genetic relationships with MEMO1. We experimentally confirmed several interactions between MEMO1 and iron-related proteins in living cells, most notably, transferrin receptor 2 (*TFR2*), mitoferrin-2 (*SLC25A28*), and the global iron response regulator IRP1 (*ACO1*). These interactions indicate that cells with high-MEMO1 expression levels are hypersensitive to the disruptions in iron distribution. Our data also indicate that MEMO1 is involved in ferroptosis and is linked to iron supply to mitochondria. We have found that purified MEMO1 binds iron with high affinity under redox conditions mimicking intracellular environment and solved MEMO1 structures in complex with iron and copper. Our work reveals that the iron coordination mode in MEMO1 is very similar to that of iron-containing extradiol dioxygenases, which also display a similar structural fold. We conclude that MEMO1 is an iron-binding protein that modulates iron homeostasis in cancer cells.

## Editor's evaluation

This important work demonstrates a function for MEMO1, a poorly understood protein that is commonly dysregulated in cancer. They provide convincing evidence that MEMO1 binds iron, and interacts with a number of known proteins involved in iron metabolism such as transferrin and mitoferrin. It still remains unclear whether the downstream metabolic programs affected by this protein in cancer are directly related to its iron binding activity or other effects of it in cell metabolism, which should be the focus of future work.

## Introduction

MEMO1 is a highly conserved protein found in the cytosol of eukaryotic cells from yeast to humans. MEMO1 appears to play a crucial role in cell motility, and has been linked to several biological processes, but its primary function remains unknown. MEMO1 has been implicated in lifespan changes in *Caenorhabditis elegans* and mice (*Ewald et al., 2017*; *Haenzi et al., 2014*), regulation of vitamin D metabolism (*Moor et al., 2018a*), and bone and central nervous system development (*Moor et al., 2018b*; *Nakagawa et al., 2019*; *Van Otterloo et al., 2016*).

In cancer context, MEMO1 supports the ability of breast tumor cells to invade surrounding tissues, leading to metastasis (*MacDonald et al., 2014*; *Marone et al., 2004*; *Meira et al., 2009*). Knockdown of MEMO1 expression reduces breast cancer cell migration in culture, and significantly suppresses lung metastasis in a xenograft model (*MacDonald et al., 2014*). In the clinical setting, retrospective analysis of resected tumors showed a strong correlation between increased expression of MEMO1 and reduced patient survival (*MacDonald et al., 2014*). These effects have been linked to the interaction between the ERBB2 (HER2) receptor and MEMO1, which in turn was proposed to relay the activation of ERBB receptor heterodimers to the microtubule cytoskeleton, thus inducing growth of lamellipodia and enabling cancer cell migration (*Marone et al., 2004*). The interaction with ERBB2 gave MEMO1 its name (**m**ediator of **E**RBB2-driven cell **mo**tility **1**). MEMO1 also contributes to breast carcinogenesis through the insulin receptor substrate protein 1 pathway (*Sorokin and Chen, 2013*) and through the interaction with the extranuclear estrogen receptor (*Frei et al., 2016*; *Jiang et al., 2013*).

MEMO1 was shown to catalyze copper-dependent redox reactions, such as superoxide radical production (*MacDonald et al., 2014*). This led to the idea that MEMO1 is required for sustained reactive oxygen species (ROS) production, likely in conjunction with NADPH-oxidase 1 activation. ROS are known to modulate functions of several proteins required for cell motility (*Block and Gorin, 2012*). Alternatively, MEMO1 has been proposed to protect cells from ROS generation by sequestering copper (*Zhang et al., 2022*). At the same time, sequence homology and structure analyses revealed strong similarity of MEMO1 to iron-containing dioxygenases, redox enzymes that catalyze the incorporation of molecular oxygen into organic molecules (*Qiu et al., 2008*), suggesting that MEMO1 may have other, iron-dependent, functions in the cell. Strikingly, within the very distinct cluster of four top matches to MEMO1 structure found in the Protein Data Bank using DALI (*Holm and Laakso, 2016*), three proved to be iron dioxygenases, with the fourth being a putative dioxygenase.

Additional clues to MEMO1 function were offered by the genome-wide studies of genetic interactions (GIs) in yeast. *MEMO1* homolog in *Saccharomyces cerevisiae* is *MHO1*, with 37% amino acid residue identity. In the global network of GIs in yeast (*Costanzo et al., 2016*), the highest interaction profile similarity was found between *MHO1* and *LSO1*, which encodes a protein strongly induced in response to iron deprivation, and a likely component of the iron transport pathway in yeast (*An et al., 2015*). Among the five highest scoring hits in the profile similarity search, there was another iron-linked protein, the ferredoxin reductase ARH1, an iron-sulfur protein playing an essential role in [2Fe-2S] cluster biogenesis (*Braymer and Lill, 2017*). Thus, iron-mediated redox reactions emerged as a common MEMO1 denominator from two orthogonal bioinformatics approaches, prompting us to investigate its iron connection.

In the present work, we show that MEMO1 is an iron-binding protein involved in iron metabolism in the cell. Iron has been implicated in carcinogenesis, tumor growth, and metastasis, in particular in breast cancer, by multiple lines of evidence, due to its potentially disruptive role in redox balance in the cell and also because of elevated iron requirements of the rapidly proliferating cancer cells (*Marques et al., 2014*; *Torti and Torti, 2013*). Thus, MEMO1 emerges as a direct molecular link between iron metabolism and metastasis in breast cancer.

## Results

### MEMO1 displays GIs with multiple iron-related proteins

Our analysis of the Cancer Genome Atlas (https://www.cancer.gov/tcga) data reveals that, in addition to breast cancer ($p < 10^{-29}$ by Mann-Whitney U-test), MEMO1 is overexpressed in the malignancies of colon, lung, and uterine origins ($p < 0.0001$), among others, while kidney, head, and neck squamous cell tumors and melanoma show little or no difference in median expression levels (*Figure 1A and B*). Further analyses of MEMO1 levels in breast cancer subtypes determined that MEMO1 overexpression

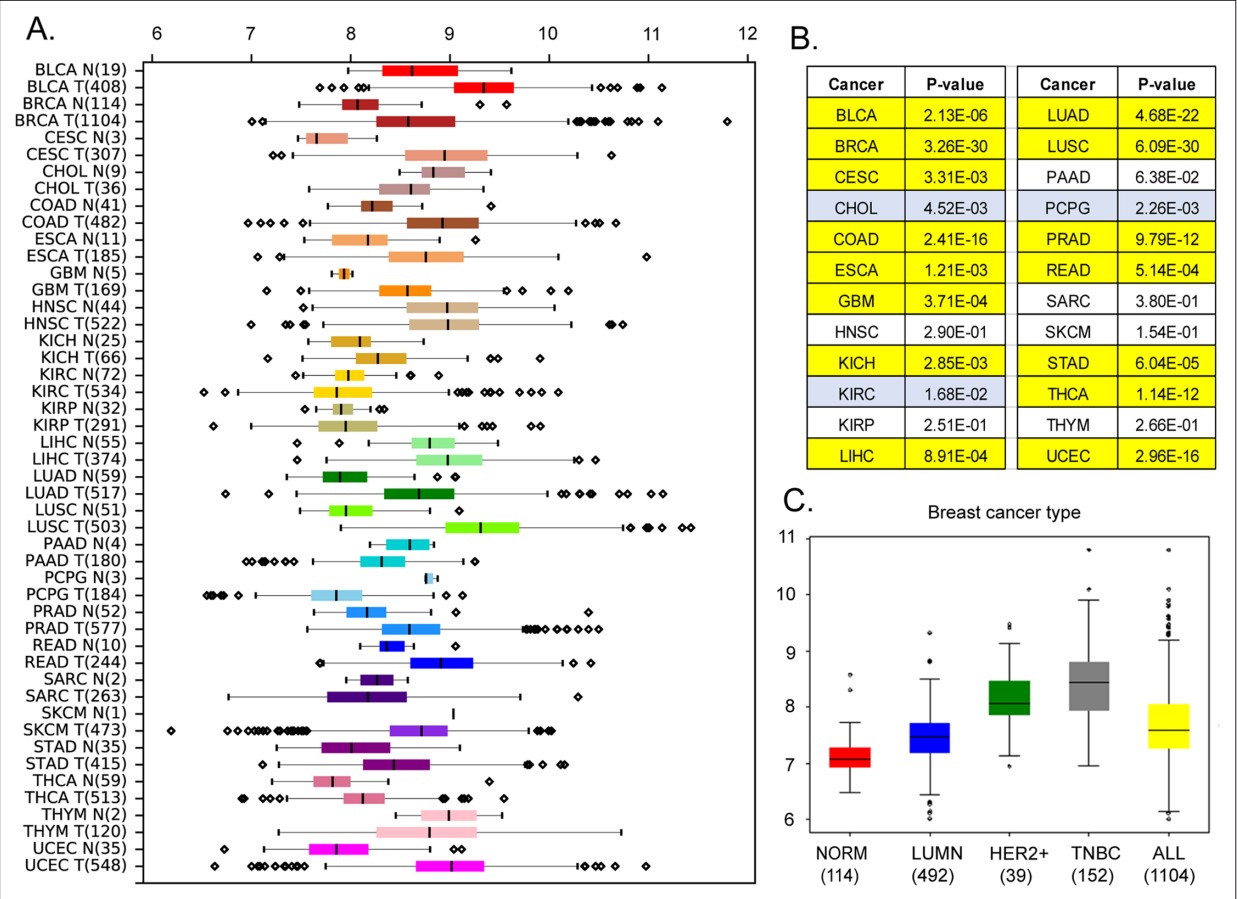

**Figure 1.** Expression levels of MEMO1 in tumors and corresponding normal tissue in various malignancies. The data is from The Cancer Genome Atlas (TCGA) database (https://www.cancer.gov/tcga). (**A**) Tissue profile of MEMO1 expression in cancer. Standard TCGA cancer type abbreviations are used (BRCA – breast cancer, SKCM – melanoma). (**B**) Statistically significant differences in MEMO1 expression levels between the tumors and the corresponding normal tissue (p<0.05 by Mann-Whitney U-test) are highlighted in *yellow* (higher expression in tumors) or *blue* (lower expression in tumors). (**C**) Subtype analysis of MEMO1 expression in breast cancer, including luminal (*blue*), HER2 positive (*green*), and triple-negative breast cancer (TNBC, *gray*). All differences vs. normal breast tissue (red) are highly significant (p<10⁻¹⁷). Note the logarithmic scale (RSEM log$_2$) on the transcript level axis in A and C.

The online version of this article includes the following source data and figure supplement(s) for figure 1:

**Source data 1.** MEMO1 expression in various cancers compared to normal tissue.

**Figure supplement 1.** Enrichment plots for MEMO1 loss-of-function genetic interaction hits.

**Figure supplement 2.** Enrichment plots for MEMO1 gain-of-function genetic interactions hits.

**Figure supplement 3.** Iron-related genes exhibiting gain-of-function (GOF) interactions with MEMO1.

**Figure supplement 4.** Iron-related genes exhibiting loss-of-function (LOF) interactions with MEMO1.

**Figure supplement 5.** MEMO1 knockout and knockdown in breast cancer and melanoma cell lines result in decrease in cellular motility and cell growth rate.

is more prominent in triple negative (TNBC) and HER2+ (HER2-enriched) breast tumors than in the luminal subtype (*Figure 1C*). This makes MEMO1 a potential therapeutic target in HER2+ breast cancer and, importantly, in TNBC, where few effective treatment options exist, and patient survival is poor. Therefore, we set out to investigate the molecular function of MEMO1.

Structural similarity of MEMO1 to the iron-containing dioxygenases and a clear link between MEMO1 homolog MHO1 and iron metabolism that emerged from genome-wide analyses of GIs networks in yeast led us to a hypothesis that MEMO1 plays a role in iron metabolism in cancer cells. As GIs are known to be functionally coherent (*Parameswaran et al., 2019*), we applied a novel approach to identify GIs of MEMO1 using publicly available gene essentiality data from multiple cell lines. We called a given gene to exhibit GI with MEMO1 if this gene was found to become essential only in

cancer cells that either overexpress or downregulate MEMO1. Using previously published genome-wide screens, we analyzed gene essentiality scores derived from 1028 cancer cell lines (*Cheung et al., 2011*; *Cowley et al., 2014*; *Kryukov et al., 2016*; *Marcotte et al., 2012*; *McDonald et al., 2017*; *Meyers et al., 2017*; *Tsherniak et al., 2017*) displaying a wide range of MEMO1 expression levels.

This analysis yielded genes that become highly essential (Wilcoxon rank-sum test p<0.05), when MEMO1 is either under-expressed, representing GIs identified from loss of function (LOF-GIs), or over-expressed, representing GIs identified from gain of function, (GOF-GIs). Since most of the previous work on MEMO1 and its role in cancer was done in breast cancer cell lines, we analyzed the breast cancer and pan-cancer datasets separately.

Consistent with the known roles of MEMO1 (*Schotanus and Van Otterloo, 2020*), we found its GIs to be enriched in the following gene sets: cell adhesion molecule binding (false discovery rate [FDR] 1.13E-03), response to insulin (FDR 2.07E-03), insulin signaling pathway (FDR 6.83E-03), cellular response to chemical stress (FDR 2.05E-04), ERBB signaling pathway (FDR 3.47E-04), and focal adhesion (FDR 1.99E-02) (*Figure 1—figure supplement 1*) for LOF-GIs, while gene set enrichment analysis (GSEA) of GOF-GIs identified signaling by ERBB2 (FDR <1.00E-05), microtubule cytoskeleton organization (FDR 8.29E-04), extranuclear estrogen signaling (FDR 6.13E-04) (*Figure 1—figure supplement 2*). Strong enrichment of the gene sets directly related to the known MEMO1 functions convincingly validated our approach.

Remarkably, GSEA of the GOF-GIs identified multiple partially overlapping gene sets related to the mitochondrial energy metabolism and redox balance in the cell, including mitochondrial transport (FDR, 5.64E-04), mitochondrial electron transport NADH to ubiquinone (FDR 8.92E-04), respiratory electron transport chain (FDR 1.54E-02), citric acid (TCA) cycle and respiratory electron transport (FDR 5.99E-03), oxidative phosphorylation (FDR 4.32E-02), glutathione metabolism (FDR 1.45E-01), and oxidative stress induced senescence (FDR 1.70E-02) (*Figure 1—figure supplement 2*). These enrichment categories include multiple genes encoding iron containing proteins, as well as those involved in iron-dependent cellular processes, such as ferroptosis. Following this lead, we asked if enrichment of these gene sets in the MEMO1 network of the GOF-GIs may reflect specifically high sensitivity of high-MEMO1 cancer cells to the disruptions of iron metabolism.

Further analysis identified 18 genes encoding proteins involved in iron metabolism and iron transport that showed statistically significant GIs with *MEMO1* (*Supplementary file 1*, *Figure 1—figure supplements 3 and 4*). Eleven genes were found to be more essential in cancer cell lines with high expression of MEMO1 (GOF-GIs, *Supplementary file 1A*), six genes were more essential in the cell lines with low-MEMO1 levels (LOF-GIs, *Supplementary file 1B*), and one (*FTH1*) showed inconsistent types of interactions between different databases. For example, knockout or knockdown of transferrin receptor 2 (TFR2), iron transporter mitoferrin-2 (SLC25A28), or iron response protein (ACO1) selectively suppresses proliferation of the high-MEMO1 cell lines, while knockout of iron transporter DMT1 (SLC11A2) leads to a stronger inhibition of proliferation in the cell lines with low-MEMO1 expression. Other iron-related genes showing GIs with *MEMO1* encode proteins that are involved in iron-sulfur cluster biogenesis (*HSPA9, ISCU, FXN, ISCA2, NUBPL, BOLA2*), contain iron-sulfur clusters (*ACO2, LIAS*), or are involved in heme synthesis (*HMOX1*). As these results strongly support a link between MEMO1 and iron-related proteins, we next explored several of these interactions in more detail.

## MEMO1 regulates iron homeostasis in the cell

To confirm the functional link between iron homeostasis and MEMO1 experimentally, we generated clonal MEMO1 knockdowns and knockouts in MDA-MB-231 triple negative breast cancer and A-375 melanoma cell lines using the CRISPR-Cas9 technology and compared the effects of shRNA knockdown of selected genes involved in iron homeostasis on the proliferation rates of the cells with high (parental), low (knockdown), and no (knockout) MEMO1 expression. We chose a melanoma cell line for comparison, because, like breast cancer, it showed a high level of MEMO1 expression, but, unlike breast cancer, there is no statistically significant difference between MEMO1 expression levels in melanoma and normal skin tissue (*Figure 1B*). Thus, melanoma serves as a useful benchmark for a cancer, where MEMO1 overexpression is not required to support malignant transformation. In agreement with the previous reports (*MacDonald et al., 2014*; *Marone et al., 2004*), we found that MEMO1 knockout in breast cancer cells results in the loss of cell motility as assessed by the wound healing assay (*Figure 1—figure supplement 5A*). MEMO1 knockdown and knockout also decreased overall

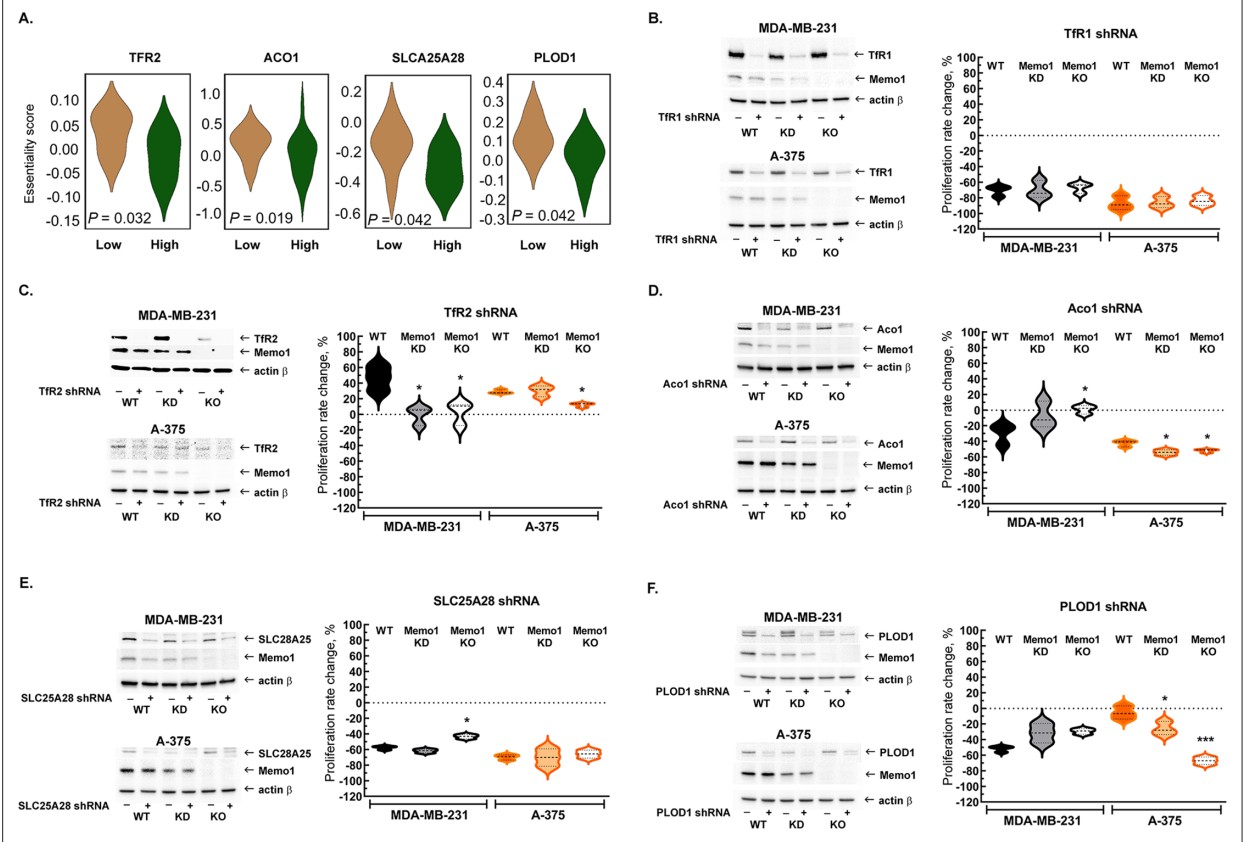

**Figure 2.** Interactions between *MEMO1* and iron-related genes. (**A**) Gene essentiality score distribution for the selected genes in the high- and low-MEMO1 expressing groups in multiple cancer cell lines as shown by the database analysis (***Supplementary file 1A***). A negative essentiality score indicates decreased cell proliferation with the gene knockdown or knockout compared to the control, a positive score indicates no effect. Effects of the TFR1 (**B**), TFR2 (**C**), ACO1 (**D**), SLC25A28 (**E**), and PLOD1 (**F**) shRNA knockdowns at different MEMO1 expression levels (WT – parental cell line, KD – *MEMO1* knockdown, KO – *MEMO1* knockout). Protein levels were detected by western blot (*left panels*). The relative shRNA effect on cell proliferation (*right panels*) is expressed as the difference between the proliferation rates with the shRNA targeting the gene of interest and the control shRNA (against the red fluorescent protein) divided by the proliferation rate with the control shRNA. Statistically significant differences vs. WT are marked (* for p<0.05 and *** for p<0.001).

The online version of this article includes the following figure supplement(s) for figure 2:

**Figure supplement 1.** Proliferation of TFR1 and TFR2 knockdowns cell lines.

**Figure supplement 2.** Proliferation of SLC25A28 knockdown cell lines.

**Figure supplement 3.** Proliferation of PLOD1 knockdown cell lines.

**Figure supplement 4.** Proliferation of HSPA9 knockdown cell lines.

**Figure supplement 5.** Proliferation of BOLA2 and NCOA4 knockdown cell lines.

rates of breast cancer cell proliferation (***Figure 1—figure supplement 5B***). As expected, no strong motility or proliferation rate dependence on MEMO1 expression level was observed in the melanoma cells (***Figure 1—figure supplement 5C and D***).

Because MEMO1 GIs identified by the gene essentiality analysis from 1028 cell lines clearly suggested a link between MEMO1 and iron transport, we started by investigating the effect of suppressing transferrin receptors TFR1 and TFR2. TFR1 is the key receptor essential for iron uptake, while TFR2 appears to be involved in the regulation of iron homeostasis rather than performing bulk iron uptake (***Kawabata, 2019***). TFR2, but not TFR1, was found to exhibit GI with MEMO1, along with other iron-related genes (***Figure 2A***). The shRNA targeting *TFR1* decreased TFR1 protein expression by more than 90% and caused near-complete proliferation inhibition in all the tested cell lines regardless of MEMO1 expression level (***Figure 2B***, ***Figure 2—figure supplement 1A and B***). Thus, TFR1

appears to be an essential protein as its loss is nearly lethal on its own, explaining why *TFR1* was not detected in our in silico screening of MEMO1 GIs.

In contrast to *TFR1*, downregulation of TFR2 by shRNA in the high-MEMO1 breast cancer cell line results in the *activation* of cell proliferation. By comparison, breast cancer cells with MEMO1 knockout or knockdown show no proliferation rate change in response to TFR2 shRNA knockdown (*Figure 2C*, *Figure 2—figure supplement 1C*), indicating that TFR2 has a GOF-GI with MEMO1. A similar MEMO1-dependent growth activation pattern was also observed in melanoma cell lines (*Figure 2C*, *Figure 2—figure supplement 1D*).

We have also experimentally tested GIs between *MEMO1* and several other iron-related genes detected in our in silico analyses (*Figure 2A*) and chosen for their key role in the regulation of iron homeostasis in the cell (*ACO1*), iron transport and processing in mitochondria (*SLC25A28* and *HSPA9*), or a direct link between the known MEMO1 function in modulating cancer cell motility and iron (*PLOD1*). *ACO1, SLC25A28,* and *PLOD1* showed GOF interactions with MEMO1 in breast cancer cells. Remarkably, *ACO1* knockdown only inhibited proliferation of the high-MEMO1 cells, but not MEMO1 knockout breast cancer cells (*Figure 2D*), while *SLC25A28* (*Figure 2E*, *Figure 2—figure supplement 2*) and *PLOD1* (*Figure 2F*, *Figure 2—figure supplement 3*) knockdowns showed some effect in all cell lines, but a stronger inhibition in the high-MEMO1 cells than in MEMO1 knockout. By comparison, none of the three genes showed GOF interactions in melanoma cells (*Figure 2D–F*,

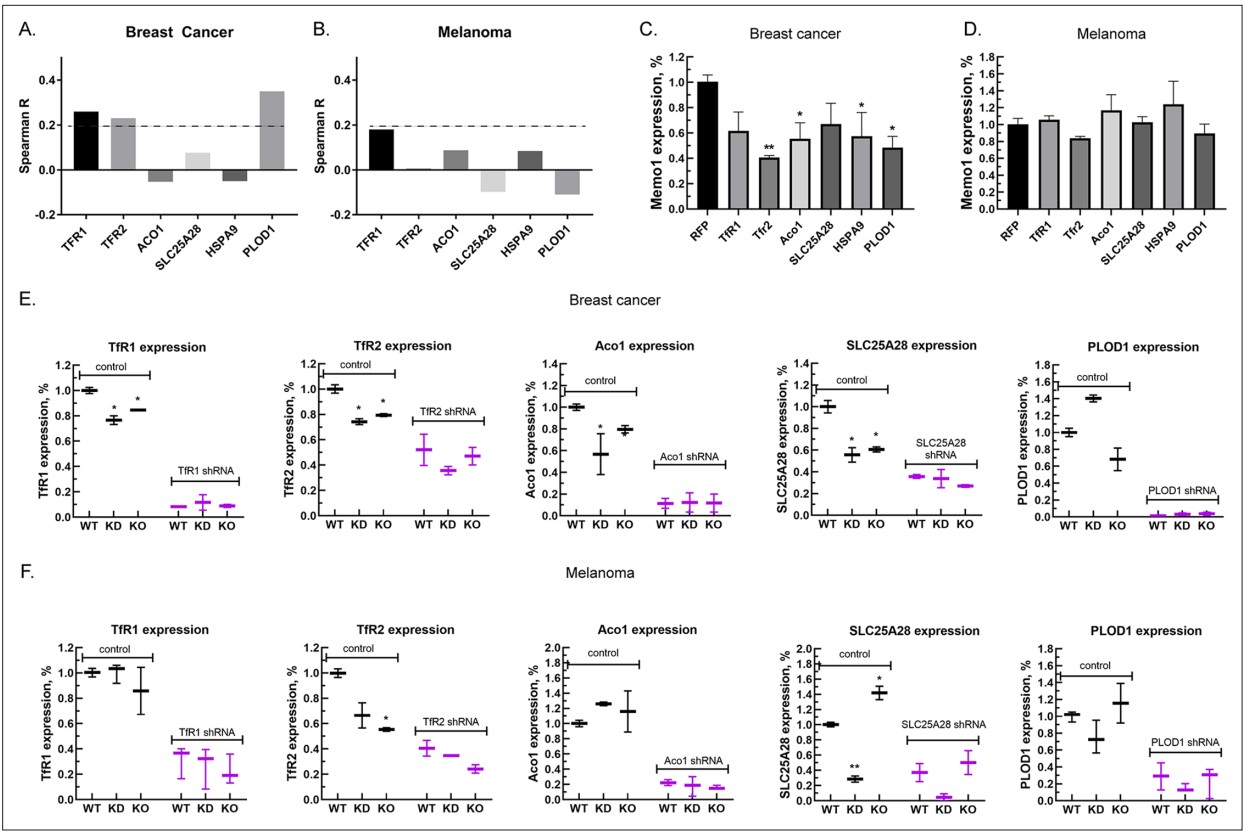

**Figure 3.** Relationship between the expression levels of MEMO1 and iron-related proteins in breast cancer and melanoma. (**A, B**) Correlations between the expression levels of MEMO1 and selected iron-related proteins in multiple breast cancer (**A**) and melanoma (**B**) cell lines ( *Figure 3—figure supplement 1*) as measured by Spearman's rank-order correlation coefficient. The dashed line indicates R=0.2 (**C, D**). Effect of the selected iron-related shRNA gene knockdowns on MEMO1 levels in breast cancer (**C**) and melanoma (**D**) cell lines detected by western blot analysis (*Figure 2B–F*). (**E, F**) Levels of the selected iron-related proteins at various MEMO1 expression levels in breast cancer (**E**) and melanoma (**F**) detected by western blot analysis in control (shRNA against the red fluorescent protein) and with the shRNA targeting the gene of interest. The data in panels C–F includes the representative blots shown in *Figure 2B–F*. Statistically significant differences vs. WT are marked (* for p<0.05 and ** for p<0.01).

The online version of this article includes the following figure supplement(s) for figure 3:

**Figure supplement 1.** Correlation between the expression levels of MEMO1 and other iron-related proteins in breast cancer (**A**) and melanoma (**B**) cell lines.

*Figure 2—figure supplements 2 and 3*). In fact, *ACO1* and *PLOD1* displayed LOF interactions in melanoma. Like TFR1, shRNA knockdown of HSPA9 results in a major proliferation inhibition of breast cancer and, particularly, of melanoma cells, regardless of MEMO1 expression level (*Figure 2—figure supplement 4*). Additionally, iron-sulfur cluster assembly factor BOLA2 showed weak but statistically significant GOF interactions with MEMO1, consistent with the results of our database analysis of gene essentiality, while knockdown of NCOA4, which modulates ferritinophagy and ferroptosis, showed some proliferation inhibition in all the cell lines, but the effect had no clear correlation with MEMO1 expression level (*Figure 2—figure supplement 5*).

Next, we investigated correlations between the expression levels of MEMO1 and the six iron-related proteins whose GIs with MEMO1 had been confirmed. Analysis of the gene expression levels in multiple breast cancer cell lines (*Figure 3A*, *Figure 3—figure supplement 1A*) revealed weak but statistically highly significant correlations between the levels of MEMO1 and TFR1 (p=4.6E-18), TFR2 (p=1.8E-14), and PLOD1 (p=1.8E-33). By comparison, none of the correlations in melanoma exceeded Spearman's rank correlation coefficient of 0.2 (*Figure 3A*, *Figure 3—figure supplement 1B*), a common, if arbitrary, correlation detection threshold.

Experimentally, knocking down TFR1, TFR2, ACO1, HSPA9, and PLOD1 in the breast cancer cell line MDA-MB-231 resulted in a decrease in the expression level of MEMO1 by 40–60% (*Figure 3C*), but, as expected, no significant effect in the melanoma cell line A-375 was observed (*Figure 3D*). Conversely, knockdown or knockout of MEMO1 in breast cancer cells resulted in a statistically significant decrease in the expression levels of TFR1, TFR2, ACO1, and SLC25A28 (*Figure 3E*). By comparison, MEMO1 knockdown or knockout in melanoma cells decreased only the expression level of TFR2 (*Figure 3F*), the effect on SLC25A28 knockdown being difficult to interpret. Thus, expression levels of MEMO1 and the other iron-related proteins are correlated in MDA-MB-231 cell line suggesting regulation through a common mechanism.

## MEMO1 plays an important role in the maintenance of iron concentration in mitochondria

GIs and expression level correlations between MEMO1 and the iron transport and iron regulating proteins suggested that MEMO1 may also be involved in regulating iron levels in the cell. To test this hypothesis, we measured iron concentration in the cytosol and in the crude mitochondrial fractions of MDA-MB-231 cells in comparison to the same fractions of MEMO1 knockdown and knockout cell lines (M67-2 and M67-9). Cells with MEMO1 knockout showed significantly lower iron concentrations in both the cytosolic (*Figure 4A*) and the mitochondrial (*Figure 4B*) fractions. The levels of iron storage protein ferritin were decreased by 30–50% in MEMO1 knockouts, both at basal iron level in the growth medium and at 20 µM added iron (*Figure 4—figure supplement 1A, B, E, F*). These results suggest that overexpression of MEMO1 allows cells to accumulate higher levels of iron, in response to higher iron demand to maintain the accelerated cell proliferation (*Figure 1—figure supplement 5B*), consistent with the increase in TFR1 levels (*Figure 3E*).

At the same time, high-MEMO1 cells were more sensitive both to the toxic effects of very high iron levels in the growth medium (*Figure 4C*), and to extreme iron deprivation caused by the high concentrations of iron chelators deferiprone and deferoxamine (DFX) (*Figure 4—figure supplement 2A and B*) than the MEMO1 knockout cells, while the cell motility was not affected by high subtoxic concentrations of iron chelators that were adjusted to create iron deficit without major effect on cell viability (*Figure 4—figure supplement 2C and D*).

The decrease in mitochondrial iron in MEMO1 knockout cells taken together with the interactions between MEMO1 and the mitochondrial iron transporter SLC25A28 (mitoferrin-2) prompted us to investigate how MEMO1 knockout affects mitochondria. We evaluated mitochondrial morphology in MDA-MB-231 cells with and without MEMO1 knockout in the basal medium and in the presence of 1 µM DFX, a high subtoxic concentration of the chelator that does not yet significantly affect cell viability (*Figure 4—figure supplement 2*), using mitochondrial marker GRP75 (HSPA9) and the trans-membrane electric potential-sensitive dye MitoTracker CM-H2Ros (*Figure 4D*). Under basal culture conditions, both the wild type and MEMO1 knockout cell lines display normal mitochondrial shape. By comparison, cells incubated overnight with 1 µM DFX present normal distribution of mitochondria in the wild type cells, while cells with MEMO1 knockout (M67-9) show perinuclear mitochondrial clustering, as detected both by GRP75 distribution and by MitoTracker CM-H$_2$Ros staining. This

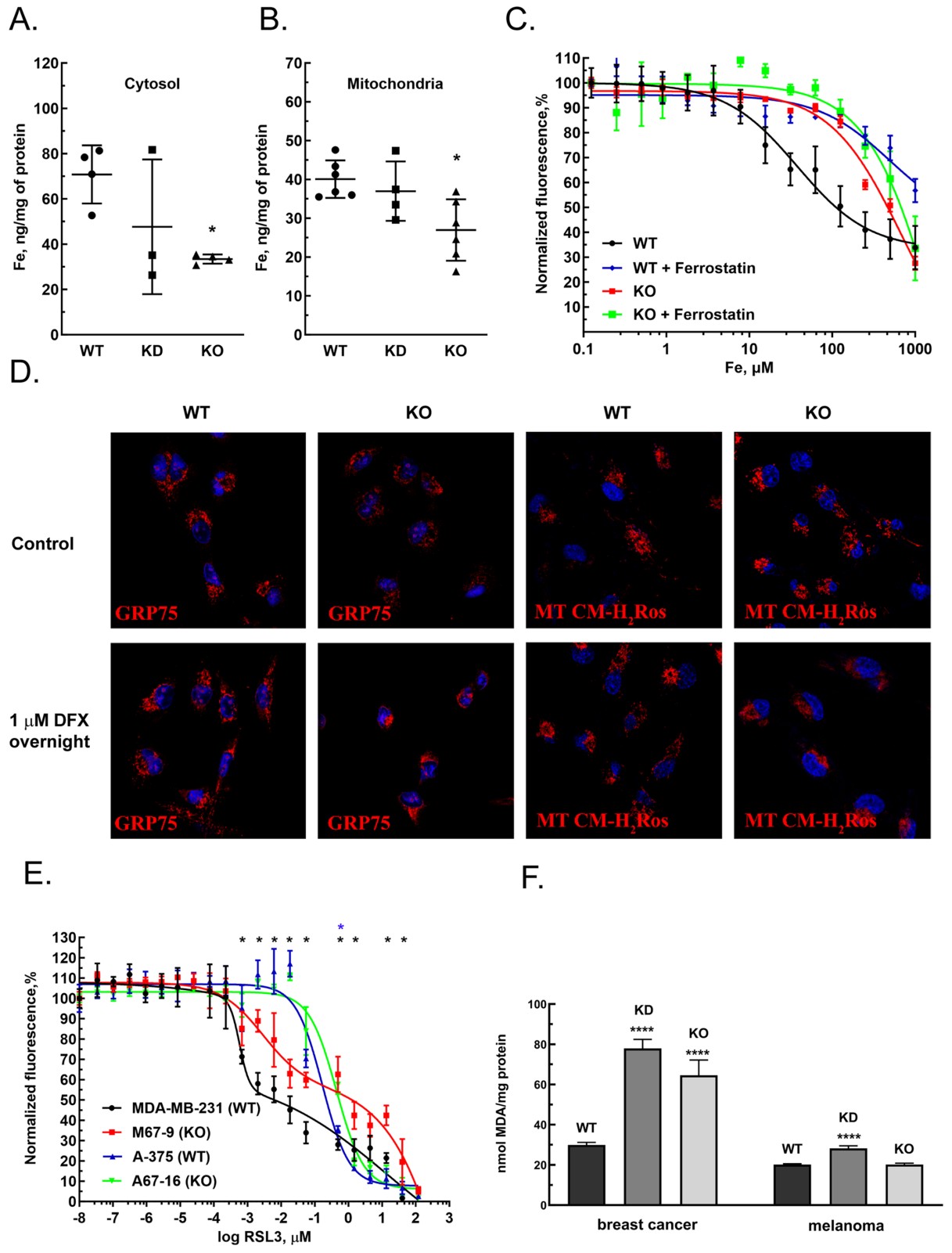

**Figure 4.** MEMO1 expression affects iron levels, mitochondrial morphology, and ferroptosis sensitivity of the cells. (**A, B**) Iron levels in the cytosolic (**A**) and mitochondrial (**B**) fractions from breast cancer cells with high (parental, WT), low (knockdown, KD), and no (knockout, KO) MEMO1 expression. Statistically significant differences vs. WT are marked (* for p<0.05). (**C**) Cell viability, measured by resazurin fluorescence as a function of iron concentration, added as ferric citrate, with and without 10 μM ferrostatin-1. (**D**) MEMO1 knockout in breast cancer cells M67-9 results in perinuclear

*Figure 4 continued on next page*

*Figure 4 continued*

mitochondrial clustering in the presence of iron chelator deferoxamine (DFX). Mitochondrial marker Grp75 (HSPA9) and MitoTracker CM-H$_2$Ros are *red*, DAPI-stained nuclei are *blue*. (**E**) MEMO1 knockout in breast cancer (M67-9) increases resistance to the ferroptosis inducer RSL3, compared to the parental cell line MDA-MB-231. RSL3 concentrations showing statistically significant difference (p<0.05) are marked with a black (breast cancer) or blue (melanoma) asterisk. (**F**) Malondialdehyde assay shows lower rates of lipid oxidation in the breast cancer cells with high-MEMO1 expression (WT) compared to cells with MEMO1 knockout (KO) and knockdown (KD) cells. The difference is smaller in melanoma cells. Statistically significant differences vs. WT (p<0.0001) are marked (****).

The online version of this article includes the following figure supplement(s) for figure 4:

**Figure supplement 1.** Expression levels of glutathione peroxidase 4 (GPX4) (**A–D**) and ferritin (**A, B, E, F**) in the breast cancer cells (**A,C,E**) and in melanoma cells (**B, D, F**).

**Figure supplement 2.** The effect of iron chelators deferoxamine (DFX) and deferiprone (DFR) on cell viability and motility.

---

observation indicates that depletion of labile iron pool caused by the incubation with DFX results in changes in mitochondrial morphology, but only in the cells without MEMO1. Thus, MEMO1 not only plays an important role in overall iron homeostasis in the cell but is specifically involved in iron regulation in mitochondria.

## MEMO1 promotes ferroptosis via increase in iron concentration in the cell

As described above, we found that MEMO1 is involved in the regulation of iron concentration in the cells. Elevated iron levels are known to make cells more susceptible to ferroptosis, a type of programmed cell death triggered by the iron-dependent lipid oxidation, and distinct from apoptosis, necrosis, autophagy, and other types of cell death (*Dixon et al., 2012*). Further suggesting a possible involvement of MEMO1 in ferroptosis, we found GOF-GI between MEMO1 and most of the proteins involved in mitochondria-based ferroptosis (*Wang et al., 2020*), including aconitase, VDACs, mitoferrin-2 (SLC25A28), frataxin (FXN), citrate synthase, Acyl-CoA synthetase long-chain family member 4, sterol carrier protein 2, sirtuin 3, glutaminase 2, SCO2, and fumarate hydratase (*Supplementary file 1C*).

Ferrostatin-1, an inhibitor of ferroptosis, significantly increased resistance of high-MEMO1 breast cancer cells to high levels of iron in the growth medium (*Figure 4C*). The IC$_{50}$ for iron was found to be 38 µM (95% confidence level 25–58 µM) without ferrostatin, compared to the IC$_{50}$ of 495 µM (95% confidence level 180–1360 µM) in the presence of ferrostatin. This effect was clearly MEMO1 dependent, as MEMO1 knockout cells were much more resistant to iron even without ferrostatin (IC$_{50}$=700 µM, 95% confidence level 494–993 µM), and ferrostatin only modestly increased their resistance (IC$_{50}$=2300 µM, 95% confidence level 449–11,900 µM). Thus, increased iron toxicity in high-MEMO1 cells is due to MEMO1 role in modulating ferroptosis, which is consistent with higher sensitivity of high-MEMO1 cells to the ferroptosis inducer RSL3 (*Figure 4E*).

To further probe the role of MEMO1 in ferroptosis, we compared responses of the wild type and MEMO1 knockout MDA-MB-231 and A-375 cells to the ferroptosis-inducing agent RSL3 (*Dixon et al., 2012*), a specific inhibitor of glutathione peroxidase 4 (GPX4), the enzyme that resides in mitochondria and plays an essential role in protecting cells against lipid oxidation (*Ingold et al., 2018*). MEMO1 knockout results in a decreased cytotoxicity of RSL3 in breast cancer cells (*Figure 4E*), indicating that high-MEMO1 cells are more sensitive to ferroptosis. Under the basal iron conditions, the GPX4 level in high-MEMO1 cells was approximately twice higher compared to the knockdown and knockout, both in breast cancer and in melanoma cells, but at the elevated iron level in the growth medium, GPX4 expression in high-MEMO1 cells decreased and no consistent difference was observed between the high- and low-MEMO1 cells (*Figure 4—figure supplement 2A–D*). While the mechanism of this decrease is unclear, this result indicates that higher sensitivity of high-MEMO1 cells to ferroptosis is due to the MEMO1-dependent differences in iron level or distribution in the cell, and not due to the lower GPX4 expression.

Lipid oxidation is one of the hallmarks of ferroptosis. Breast cancer cells with MEMO1 knockdown and knockout have significantly higher levels of lipid oxidation product malondialdehyde (MDA) (*Figure 4F*). Taken together, these results suggest that MEMO1 knockout increases ferroptosis resistance in breast cancer cells through the reduced iron pool (*Figure 4A and B*) and a higher tolerance toward lipid oxidation products. Melanoma cells show much smaller MEMO1-dependent variations in

the MDA level than breast cancer cells, consistent with the observed smaller difference in RSL3 sensitivity between the wild type and MEMO1 knockout in melanoma (*Figure 4E*). Overall, these results indicate that MEMO1 overexpression sensitizes breast cancer cells to ferroptosis.

## MEMO1 is an iron-binding protein

We have discovered GIs between MEMO1 and many iron-dependent proteins and found strong indications of MEMO1 involvement in the regulation of iron levels in the cell and in ferroptosis. On the other hand, previous work revealed a putative metal-binding site in the structure of MEMO1 (*Qiu et al., 2008*) and demonstrated that it has copper-dependent redox activity (*MacDonald et al., 2014*). Copper binding to MEMO1 has been demonstrated recently (*Zhang et al., 2022*). Taken together, these findings prompted us to investigate metal-binding properties of MEMO1. We expressed MEMO1 as a fusion with the chitin-binding domain and intein and purified the protein by chitin affinity chromatography combined with intein self-cleavage, followed by size exclusion chromatography, producing highly pure, well-folded protein, as evidenced by NMR spectroscopy (*Figure 5—figure supplement 1*).

Next, we investigated metal binding to pure MEMO1, in the presence of a mixture of the reduced (GSH) and oxidized (GSSG) glutathione (9.5 mM GSH and 0.5 mM GSSG), approximating redox environment in the cytosol (*Deponte, 2013*). Inductively coupled plasma mass-spectrometry (ICP-MS) showed that, under these conditions, MEMO1 binds stoichiometric amounts of iron (approximately 1:1 molar ratio). In the absence of iron, we have also observed binding of copper (added as Cu(I)) to MEMO1 at a molar ratio of copper to protein of approximately 0.4. Under oxidative conditions, without glutathione, or other reducing agents, copper, added as Cu(II), bound to MEMO1 at approximately 0.7 molar ratio.

To determine the metal-binding affinity, we measured dissociation constants for iron and copper by isothermal titration calorimetry (ITC). The measured $K_d$ value for iron in the presence of glutathione was $5.0 \pm 2.6 \times 10^{-6}$ M (*Figure 5A*) consistent with the value of $2.4 \pm 1.0 \times 10^{-6}$ M determined by microscale thermophoresis (*Figure 5—figure supplement 2*). Glutathione itself binds iron with a $K_d$ of about $10^{-5}$ M (*Hider and Kong, 2011*), and therefore MEMO1 affinity for free $Fe^{2+}$ ions may be much higher. However, the apparent $K_d$ value measured in the presence of an abundant intracellular low-affinity iron acceptor, such as glutathione, is physiologically more relevant. Copper (I) binding was also detected by ITC, with an estimated $K_d$ of about $3 \times 10^{-6}$ M (*Figure 5—figure supplement 3*).

To reveal the binding mode of each metal, we have crystallized MEMO1 in the presence of iron or copper under the conditions similar to those used previously for solving the structure of the metal-free protein (*Qiu et al., 2008*). To prevent metal oxidation, MEMO1 crystallization with iron (II) or copper (I) was set up in an anaerobic chamber and the plates were incubated under argon, while the crystals were growing. We have solved the structures of MEMO1 with iron (*Figure 5B*) at the resolution of 2.15 Å and copper at 2.5 Å, by molecular replacement using the previously solved structure of the metal-free protein (PDB ID 3BCZ) (*Qiu et al., 2008*; *Supplementary file 1D*). The protein fold of both metal-bound forms of MEMO1 is essentially identical to the metal-free structure, with RMSD for α-carbons of the superimposed structures being less than 0.17 Å. In the MEMO1-Fe structure, iron density is clearly visible in the region of the previously predicted metal-binding site with an overall occupancy of 40–60%, as confirmed by the anomalous diffraction map. Iron is coordinated by H49, H81, and C244 (*Figure 5C*). As noted previously, these residues correspond to the iron-coordinating residues H12, H61, and E242 in the structure alignment of MEMO1 with catechol dioxygenase LigB (*Qiu et al., 2008*), to H12, H59, and E239 in the more recent structure of the bacterial gallate dioxygenase DesB (PDB ID: 3WR8), and to H13, H62, and E251 in the aminophenol dioxygenase from *Comamonas* sp. (PDB ID: 3VSH). Iron coordination by two histidines and an aspartate or glutamate is also observed in many other non-heme dioxygenases (*Abu-Omar et al., 2005*). Thus, iron coordination by two histidines and a cysteine in MEMO1 is an unusual variation on a common theme. A glutathione molecule was also found in the MEMO1-Fe structure, close to the iron-binding site, with the glutathione glycine carboxyl forming an electrostatic interaction with H192. This finding is consistent with Fe-GSH binding to MEMO1 observed by ITC. When MEMO1 was crystallized in the presence of copper, the copper atom bound at the same site, and was coordinated by the same residues as iron (*Figure 5D*).

To confirm the iron-binding site, we generated H49A and C244S mutants, along with H192A and D189N variants, because the latter residues are located in the proximity of the bound iron and were

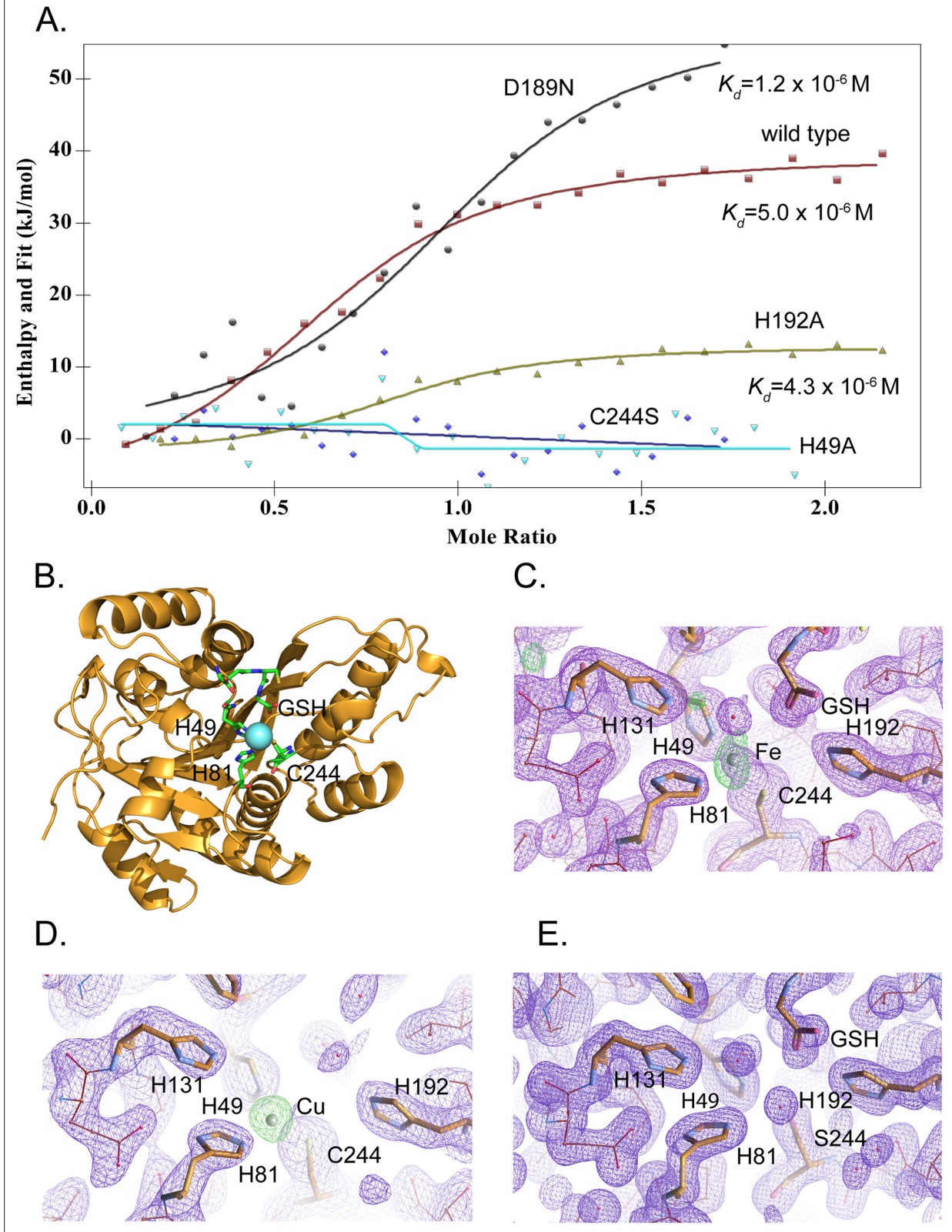

**Figure 5.** MEMO1 binds iron or copper in the site formed by H49, H81, and C244. (**A**) Iron binding to the wild type MEMO1 and metal-binding site mutants analyzed by isothermal titration calorimetry. Dissociation constant ($K_d$) values are shown under the fitted binding curves for the wild type (*squares*), D189N (*circles*), and H192A (*triangles*) MEMO1 variants. The C244S (*diamonds*) and H49A (*inverted triangles*) variants did not bind iron. (**B**) Structure of the wild type MEMO1 with iron (PDB ID 7KQ8). (**C, D**) Anomalous difference electron density maps showing iron (7KQ8) (**C**) and copper

*Figure 5 continued on next page*

Figure 5 continued

(7L5C) (**D**) coordinated by H49, H81, and C244. The H131 and H192 residues, albeit close to the metal-binding site, do not participate in metal coordination. GSH is glutathione. (**E**) Region of the electron density map of the C244S-MEMO1 (7M8H) corresponding to the metal-binding site in the wild type (cf. panels C and D).

The online version of this article includes the following figure supplement(s) for figure 5:

**Figure supplement 1.** Fingerprint $^1$H,$^{15}$N-TROSY spectra of the wild type MEMO1 and several metal-binding site mutants recorded at 900 MHz.

**Figure supplement 2.** Iron binding to the wild type MEMO1 analyzed by microscale thermophoresis.

**Figure supplement 3.** Copper binding to the wild type MEMO1 analyzed by isothermal titration calorimetry.

previously proposed to belong to the metal-binding site of the protein (**MacDonald et al., 2014**). All the mutant proteins were properly folded as shown by NMR (**Figure 5—figure supplement 1**). As shown by ITC (**Figure 5A**), iron-binding affinity in the H192A and D189N mutants is not significantly changed, whereas H49A and C244S variants do not bind iron at all. Consistent with the crucial role of C244 in metal binding, no metal density was found in the C244S variant of MEMO1 crystallized in the presence of copper (**Figure 5E**). Thus, metal coordination in MEMO1 is achieved by H49, H81, and C244, while H192 and D189 do not participate in metal binding in MEMO1.

## Discussion

In summary, our results firmly link MEMO1 to a complex network of iron-dependent processes in the cell. We have shown that MEMO1 is an iron-binding protein that exhibits GIs with many other iron-related proteins and regulates iron levels in the cell. Since MEMO1 can bind both iron and copper in vitro, the question arises, which metal is bound to MEMO1 in the cell. Our $K_d$ measurements suggest that considerations based on the equilibrium dissociation constants should apply to iron binding in the living cell. Concentration of iron in the labile, i.e., readily exchangeable pool, in the cell is within the low micromolar range (**Epsztejn et al., 1997**). Therefore, in the absence of competition with another metal, MEMO1 would bind iron from this pool, as we have demonstrated above for glutathione.

The situation is very different for copper. There is essentially no free or readily exchangeable copper in the cell, as all available copper is tightly bound to proteins with $K_d$ on the order of $10^{-13}$ M, or less, such as copper chaperones ATOX1 and CCS, metal-binding domains of copper ATPases ATP7A and ATP7B, and others (**Banci et al., 2010**). Copper transfer between proteins in the cell requires specific protein-protein interactions and is kinetically limited by the rate of such interactions. Therefore, in the absence of specific copper loading mechanisms for MEMO1, it is likely to predominantly bind iron in the cytosol of the living cells, even though copper binding can be observed in vitro. Still, consistent with the previous reports (**MacDonald et al., 2014**; **Zhang et al., 2022**), MEMO1 may bind copper under oxidative conditions in a specific local environment within the cell, and it is tempting to speculate that metal-binding change may trigger a switch between different MEMO1 activities in the cell.

The iron-binding site of MEMO1 is structurally very similar to that of iron-containing extradiol dioxygenases, with a notable difference of a cysteine residue (C244) located at the position occupied by a glutamate in those proteins. No dioxygenase activity has been reported for MEMO1, and we have so far been unable to detect any with a variety of standard substrates we have tried, such as gallate, protocatechuate, 3,4-dihydroxyphenylalanine (L-DOPA), and others. Still, the remarkable structural similarity between MEMO1 and dioxygenases strongly suggests that MEMO1 may catalyze redox reactions involved in biosynthesis or breakdown of a signaling molecule in cancer cells.

We have validated GIs between *MEMO1* and many genes encoding iron-related proteins by studying the effects of LOF of those gene products on the proliferation of breast cancer and melanoma cell lines with different expression levels of MEMO1. Perhaps, the most interesting connection that emerged from these experiments is between MEMO1 and TFR2. Both transferrin receptors in human cells, TFR1 and TFR2, are involved in iron transport into the cells. However, whereas TFR1 is an essential protein that accounts for bulk iron uptake, TFR2 appears to play a regulatory role (**Kawabata, 2019**). Selective activation of high-MEMO breast cancer cell proliferation by TFR2 knockdown and a marked decrease in cytosolic iron concentration in the context of MEMO1 knockout suggest an important role for MEMO1 in maintaining iron homeostasis in cancer cells, likely in conjunction with TFR2 (**Figure 6**). MEMO1-dependent activation of cell proliferation by TFR2 knockdown, and the link

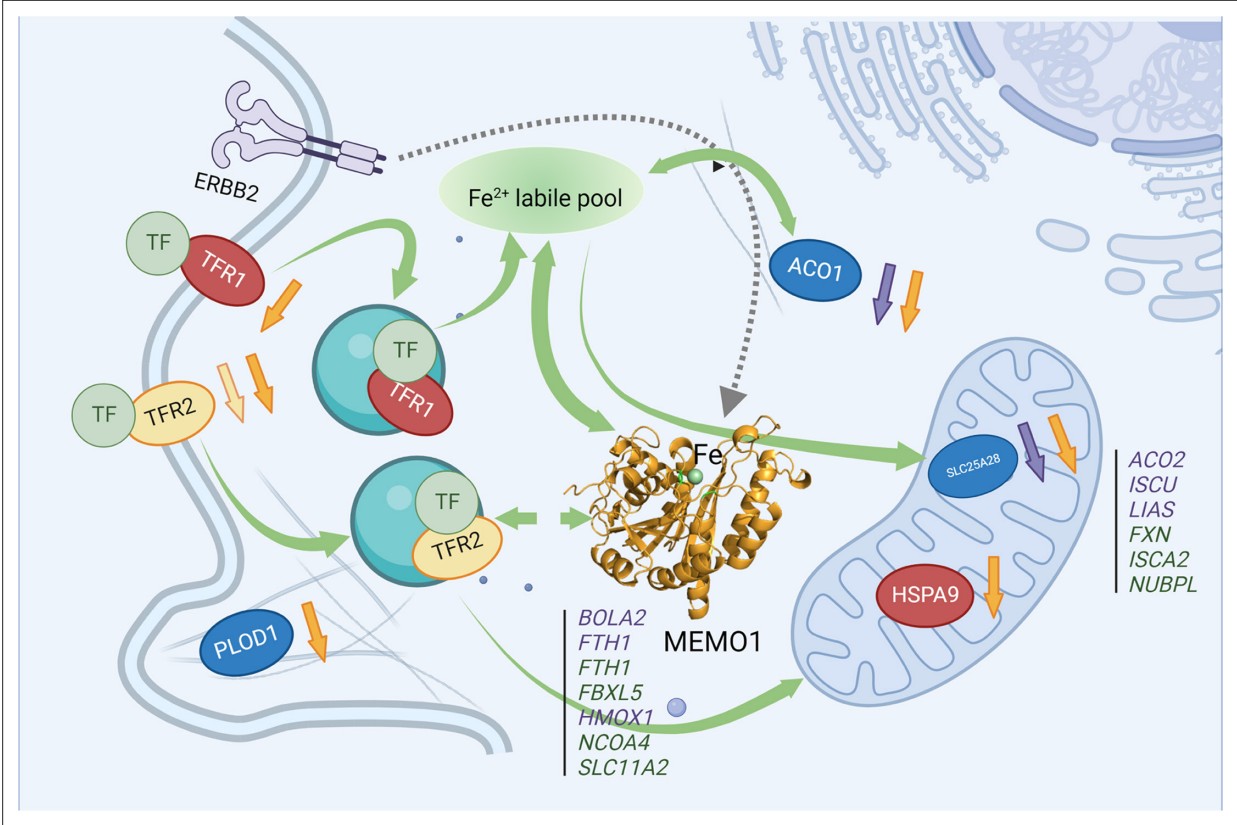

**Figure 6.** MEMO1 interactions with the other iron-related proteins in breast cancer cells. Proteins involved in the experimentally validated genetic interactions with MEMO1 in the present work are shown as ovals. Other iron-related genes showing interactions with *MEMO1* by database screening are listed in columns, separately for cytosolic or nuclear and mitochondrial proteins. Proteins or genes shown in *blue* display SDL interactions with MEMO1; those shown in *green* display SL interactions with MEMO1. In *red* are essential proteins; their knockdown severely inhibits cell proliferation regardless of MEMO1 expression status. TFR2 (*yellow*) knockdown uniquely stimulates proliferation of high-MEMO1 cells. Curved *green* unidirectional arrows show iron transport pathways. Curved *green* bidirectional arrows show iron-dependent regulation. Dashed *gray* arrow indicates MEMO1 interaction with ERBB2 receptor. Short up or down arrows indicate reciprocal effects of the MEMO1 and the interacting genes knockouts and knockdowns: protein expression change in low/no MEMO1 cells is indicated by the same color as the protein, MEMO1 expression change with the protein knockdown is indicated by the *orange* arrow. TF transferrin.

between MEMO1 and TFR2 expression levels is also observed in melanoma cells, suggesting that MEMO1-TFR2 interaction has a salient role in regulating iron in various cell types.

The link between MEMO1 and iron homeostasis is further supported by the GOF-GI between MEMO1 and the iron-dependent regulatory protein ACO1. ACO1, in its iron-free state, functions as an iron response protein (IRP1): it binds to the iron-responsive elements in the 3′-untranslated regions of mRNA encoding iron transporters TFR1 and DMT1, and in the 5′-untranslated regions of mRNA encoding several other iron-dependent proteins, including mitochondrial aconitase ACO2, iron efflux protein FPN1, and iron storage protein ferritin (FTN and FTH subunits) (*Lane et al., 2015*). Upon iron binding, IRP1 dissociates from mRNA and becomes a functional cytosolic aconitate hydratase (aconitase). IRP1 dissociation results in the destabilization of TFR1 mRNA and a decrease in iron uptake (*Rouault et al., 1991*).

The selective suppression of the high-MEMO1 cells proliferation by ACO1 knockdown indicates dysregulation of MEMO1-dependent aspects of iron homeostasis in the cell. One of the possible explanations for this finding is a need for a tighter control of iron homeostasis in high-MEMO1 and high-iron cells compared with the cells expressing less MEMO1 and containing less iron. The link between MEMO1 and TFR2 and ACO1 would suggest a possible interaction between MEMO1 and TFR1. However, TFR1 knockdown strongly inhibited proliferation of all cell lines, regardless of MEMO1

expression levels, consistent with the essential role of TFR1 in iron uptake. Several other genes regulated by ACO1, including DMT1, FTH1, and ACO2, were detected in our database screening, indicating that MEMO1 is an integral part of iron regulatory network of GIs in the cell.

The MEMO1 iron connection trail leads to mitochondria, as the major intracellular iron consuming organelle. Iron is a cofactor of the highly abundant electron transport proteins in the respiratory chain and several tricarboxylic acid cycle enzymes. It is, therefore, logical to expect that a disruption of iron homeostasis caused by MEMO1 knockout will affect mitochondrial functions. Indeed, we observed that an iron chelator, DFX, causes major changes in the mitochondrial morphology of MEMO1 knockout cells, at a concentration that does not affect the high-MEMO1 parental cells. Perinuclear mitochondrial clustering is one of manifestations of hypoxia (*Agarwal and Ganesh, 2020*), which, in turn, may be caused by iron deficiency (*Lane et al., 2015*; *Renassia and Peyssonnaux, 2019*). Absence of the perinuclear clustering in the mitochondria of high-MEMO1 cells combined with higher iron level compared to MEMO1 knockouts suggests that MEMO1 specifically facilitates iron transport into mitochondria. MEMO1 overexpression may help to maintain normal metabolism of cancer cells by increasing iron levels in mitochondrial under hypoxic conditions that are frequently found in tumors.

Remarkably, TFR2 has been previously reported to be a part of the novel iron transport pathway to mitochondria in substantia nigra dopamine neurons (*Mastroberardino et al., 2009*) and in erythroid progenitor cells (*Khalil et al., 2017*). Although it remains to be determined, whether a similar pathway exists in breast cancer cells that we studied, our data provide the first indication that MEMO1 and TFR2 may interact in the iron transport to mitochondria.

Several proteins involved in the biosynthesis of iron-containing cofactors in mitochondria showed GIs with MEMO1 in our genome-wide in silico screening. SLC25A28 (mitoferrin-2), which mediates iron uptake in mitochondria, displayed GOF-GI with MEMO1, its knockdown selectively suppressing proliferation of MEMO1-overexpressing breast cancer cells. This may indicate that dysregulation of iron transport processes in high-MEMO1 cells makes them more susceptible to ferroptosis by diverting iron off pathway, to participate in the damaging oxidative reactions. In fact, in our experiments, high-MEMO1 breast cancer cells were significantly more sensitive to the ferroptosis inducer RSL3 and to increased iron concentrations in the medium than MEMO1 knockouts.

GIs between MEMO1 and PLOD1 in breast cancer cells that we found are particularly notable. PLOD1 is highly expressed in many malignant tumors, likely contributing to the epithelial to mesenchymal cell transition in the course of cancer progression (*Wang et al., 2021*). The MEMO1-PLOD1 GI establishes a direct connection between iron, MEMO1, and cancer cell motility. PLOD1 is an iron-containing enzyme, which participates in the collagen assembly and in the regulation of collagen synthesis and extracellular matrix remodeling, consistent with the established role of MEMO1 in cancer cell tissue invasion and metastasis.

Many of the GIs between MEMO1 and the other iron-related proteins that we have investigated in the present work manifest themselves not only in MEMO1-dependent effects of the second gene knockdown on cell proliferation, but also in the connections between the expression levels of the two proteins. Such connections are revealed both in the weak but statistically highly significant correlations between the expression levels of MEMO1 and TFR1, TFR2, and PLOD1 across multiple breast cancer cell lines and in the reciprocal effects of gene knockdowns and knockouts on protein expression observed in our experiments. Thus, TFR2 and SLC25A28 (mitoferrin-2) levels are markedly decreased in breast cancer cells with MEMO1 knockdown and knockout. Conversely, TFR1, TFR2, SLC25A28, and PLOD1 knockdowns decrease expression levels of MEMO1. These correlations indicate coregulation of expression of many iron-related genes and *MEMO1* (*Figure 6*).

Across the board comparison of MEMO1 GIs and MEMO1 knockout effects between breast cancer and melanoma cells indicate that variations in MEMO1 levels overall have stronger effects in breast cancer cells than in melanoma. Therefore, MEMO1 overexpression relative to the normal tissue rather than absolute MEMO1 levels in the cell appears to be a hallmark of hypersensitivity to iron homeostasis disruption.

In summary, our work has revealed that MEMO1 is an iron-binding protein that regulates iron homeostasis in cancer cells. MEMO1 overexpression may help to maintain normal metabolism of cancer cells by increasing iron levels in mitochondrial under hypoxic conditions. Thus, MEMO1 may serve as a biomarker of tumors particularly sensitive to the therapies targeting iron metabolism in the

cell. GIs of MEMO1 may be targeted to suppress metastasis in breast cancer and other malignancies with high-MEMO1 expression level. MEMO1 structure and iron coordination mode suggest that it may be involved in the biosynthesis or processing of a signal molecule in the cell.

## Methods
### Generation of Memo1 knockout and knockdown cell lines
MDA-MB-231 (breast cancer) and A-375 (melanoma) cell lines were obtained from ATCC and cultured in Dulbecco's Modified Eagle Medium (DMEM, HyClone) supplemented with 10% fetal bovine serum (FBS, Gibco). MEMO1 knockdowns and knockouts were generated using CRISPR/Cas9 TrueGuide synthetic crRNA technology (Invitrogen) with the guiding crRNA targeting MEMO1 exons 4–6. Genomic cleavage by Cas9 was confirmed using GeneArt Genomic Cleavage Detection kit (Life Technologies) to amplify the region of genomic DNA targeted for cleavage. Individual clones containing MEMO1 knockouts and knockdowns were generated by limiting dilution and tested for MEMO1 expression by western blot. All the cell lines were re-authenticated by short tandem repeat profiling using ATCC reference database and tested negative for mycoplasma contamination.

### Western blotting
Cells were scraped, washed in PBS, resuspended in lysis buffer (50 mM HEPES, pH 7.4, 150 mM NaCl, 0.2% Triton X-100, cOmplete ULTRA Protease Inhibitor Cocktail [Roche]), and incubated on ice for 30 min. Nuclei and cell debris were separated by centrifugation at 6000×$g$ for 10 min. Protein concentration in the supernatant was measured by BCA (Pierce). Proteins were separated using 4–20% Mini-Protein TGX Precast gels (Bio-Rad) and transferred to 0.2 µm nitrocellulose membrane (Bio-Rad). Membranes were blocked with 5% Amersham ECL Blocking Reagent (GE) in PBS with 0.1% Triton X-100 (PBST) for 1 hr at room temperature, then incubated with the primary antibodies at 4°C overnight. The following antibodies were used: anti-Memo1 antibody (mouse monoclonal antibody AT1E9, sc-517412, Santa Cruz Biotechnology, dilution 1:500 in 0.5% of blocking reagent on PBST), anti-Ferritin antibody (rabbit monoclonal antibody MA5-32244, Thermo Fisher Scientific, dilution 1:1000 in 1% of blocking reagent on PBST), anti-GPX4 antibody (mouse monoclonal antibody E12, sc-166570, dilution 1:500 in 0.5% of blocking reagent on PBST), anti-PLOD1 (LLH1) antibody (mouse monoclonal antibody B-5, sc-271640, Santa Cruz Biotechnology, dilution 1:500 in 0.5% of blocking reagent on PBST), anti-TfR1 antibody (mouse monoclonal anti-CD71 antibody 3B8 2A1, Santa Cruz Biotechnology, dilution 1:500 in 0.5% of blocking reagent on PBST), anti-TfR2 antibody (mouse monoclonal antibody B-6, sc-376278, Santa Cruz Biotechnology, dilution 1:100 in 0.5% of blocking reagent on PBST), anti-Aco1 antibody (rabbit polyclonal antibody PA5-41753, Invitrogen, dilution 1:1000 in 0.5% of blocking reagent on PBST), anti-SCL25A28 (rabbit polyclonal antibody against mitoferrin 2, BS-7157R, Bioss, Thermo Fisher Scientific, dilution 1:500 in 0.5% of blocking reagent on PBST), anti-Grp75 antibody (mouse monoclonal anti-HSPA9 antibody D-9, sc-133137, Santa Cruz Biotechnology, dilution 1:1000 in 0.5% of blocking reagent on PBST), anti-actin β antibody (MA5-15452, Invitrogen, dilution 1:10,000 in 0.5% of blocking reagent on PBST). Following incubation with primary antibodies, membranes were washed in 0.5% blocking reagent on PBST three times for 5 min and incubated with the following secondary antibodies for 1 hr at room temperature: Amersham ECL anti-mouse IgG, horseradish peroxidase (HRP)-linked species-specific whole antibody (from sheep) (NA931V), dilution 1:1000, or Pierce donkey anti-rabbit HRP-linked antibody (#PI31458), dilution 1:2500. Membranes probed with anti-TfR2 antibody were incubated with m-IgGk BP-HRP secondary antibody (Santa Cruz, sc-525409, dilution 1:1000). After washing and incubation with the secondary antibody, membranes were washed with 0.5% blocking reagent in PBST (three times for 5 min) and imaged using standard ECL solutions and G:BOX (Syngene).

### Genome-wide in silico screening
Using a previously described concept (*Kryukov et al., 2016*), we have screened the *Marcotte et al., 2016*, and Achilles Project databases (*Cowley et al., 2014*), and Project DRIVE (*McDonald et al., 2017*) RNAi datasets and CERES (*Meyers et al., 2017*; *Tsherniak et al., 2017*) CRISPR-Cas9 dataset. The cell lines in each dataset were classified based on the expression of MEMO1 from Cancer Cell Line Encyclopedia (CCLE) (*Barretina et al., 2012*) database. The difference in the gene essentiality

score between the top 5% of high-MEMO1 expressing cell lines and bottom 5% of low-MEMO1 cell lines for pan-cancer and top 25% of high-MEMO1 expressing cell lines and bottom 25% of low-MEMO1 cell lines for breast cancer cell lines were calculated and ranked statistically significant hits (Wilcoxon rank-sum test p<0.05) by the difference in median values of the essentiality scores. We thus generated two types of datasets, one containing the genes essential in MEMO1-low cell lines, the other in MEMO1-high cell lines. Collectively, these datasets contained results from 1028 cancer cell lines, including 92 breast cancer cell lines. Enrichment analysis for the SL/SDL interaction partners were performed using GSEA software (*Subramanian et al., 2005*).

## The shRNA knockdown assays

MEMO1 GIs predicted by genome-wide gene knockout and knockdown database analysis were validated by measuring proliferation rates of cells with high-level MEMO1 expression (parental cell lines MDA-MB-231 and A-375), MEMO1 knockdowns (M67-2 and A67-4 respectively), and complete MEMO1 knockouts (M67-9 and A67-16 respectively) with shRNA knockdowns of PLOD1, HSPA9, SLC25A28, TFR1, TFR2, or ACO1, compared to the control RFP-shRNA. Pooled shRNAs targeting each of the tested genes were delivered into the cells by lentiviral transfection. In brief, cells were transfected with lentivirus and incubated for 24 hr, then the virus was removed, and cells were incubated with puromycin for the next 48 hr, then trypsinyzed and plated onto 96-well plates in the replicates of 8 at the density of 1000 cells per well for A-375, A67-4, and A67-16 cells, and 2000 cells per well for MDA-MB-231, M67-2, and M67-9 in the presence of puromycin. Cells were imaged using Incucyte S3 (Sartorius) every 8 hr for the next 120–140 hr. Cell confluency was measured and used to calculate proliferation rates by fitting data using Logistic Growth equation with GraphPad Prism v. 9. The rest of the cells were plated onto 100 mm plates and harvested for western blot analysis after 48 hr.

## Cytotoxicity assay

Cytotoxicity of RSL3 was measured using resazurin assay. Cells were plated onto 96-well plates at the density of 1000 cells per well for MDA-MB-231 cell line and its derivatives and 750 cells per well for A-375 cell line and its derivatives. The next day cells were titrated with RSL3 and incubated for the next 48 hr. After incubation, media was discarded and replaced by the fresh one containing 88 μM resazurin (Sigma-Aldrich). Cells were incubated overnight, and fluorescence was measured using 540 nm excitation and 590 nm emission.

## MDA assay

Cells were grown on 60 mm plates to 80% confluency, harvested and resuspended in 1 ml of cold PBS. A 100 μl volume of suspension was separated for measuring protein concentration by BCA (Pierce). Cells were pelleted at 14,000×*g* for 1 min, and lipid peroxidation was measured using MDA assay kit (Abcam) according to the manufacturer's instructions.

## Immunofluorescence microscopy

Cells were plated onto glass coverslips at the confluency of ~40–50% and incubated overnight. The next day 1 μM of DFX was added and cells were incubated overnight. Following the incubation, cells were washed with cold PBS and fixed using 50:50 mix of methanol and acetone (–20°C) for 30 s, blocked in 5% BSA in PBS overnight and incubated with anti-GRP75 antibodies (D-9, Santa Cruz Biotechnology) for 1 hr at room temperature. Cells were washed three times with PBS and incubated with secondary goat anti-rabbit antibody labeled with Alexa Fluor 633 (Invitrogen) in the dark for 1 hr at room temperature. After wash in PBS, cells were mounted onto the glass microscope slides using ProLong Diamond antifade mountant with DAPI (Invitrogen) and imaged using Leica DMi8 confocal microscope after 48 hr.

Imaging with MitoTracker CM-H2Ros was performed using live cells: after incubation with DFX the media was removed and substituted with 200 nM MitoTracker solution in DMEM (no FBS), cells were incubated for 30 min, then media was discarded, and cells were incubated for 5 min in DMEM without serum. Following that, cells were washed with PBS and incubated in 8 μM Hoechst 33422 (Invitrogen) for 10 min, rinsed with PBS, placed in Live Cells Imaging Solution (Molecular Probes) and imaged on Leica DMi8 confocal microscope within 2 hr post staining.

## ICP-MS measurements

Cells were grown on 100 mm plates (6 biological replicates), rinsed with PBS, trypsinyzed, then resuspended in 1 ml of cold PBS. A 0.1 ml volume of suspension was kept for determination by BCA (Pierce), the rest was pelleted at 14,000×$g$ for 1 min. Cell pellets were resuspended in 0.4 ml of PBS and lysed by 25 passages through a 27-gauge syringe needle. Cell lysates were centrifuged for 2 min at 1000×$g$ to remove remaining whole cells and nuclei, and the supernatant was centrifuged again at 10,300×$g$ for 10 min. Supernatant containing the cytosolic fraction was collected and centrifuged for 30 min at 100,000×$g$. Pellet containing crude mitochondrial fraction was resuspended in 0.5 ml of PBS and centrifuged again, then resuspended in 50 µl PBS. Protein concentration in the samples was determined by BCA assay (Pierce). Prior to ICP-MS analysis, the cytosolic fraction was diluted with 1% nitric acid (trace metal grade, Thermo Fisher Scientific). The mitochondrial fraction was briefly digested with concentrated nitric acid (trace metal grade, Thermo Fisher Scientific) at 90°C and subsequently diluted with 1% nitric acid.

ICP-MS measurements were performed using an Agilent 7700× equipped with an ASX 500 autosampler in the OHSU Elemental Analysis Shared Resource (Oregon, USA). Data were quantified using weighed, serial dilutions of a multi-element standard (CEM 2, (VHG Labs VHG-SM70B-100) Fe, Cu, Zn), and a single element standard for Ca (inorganic ventures CGCA1) and P (VHG Labs, PPN-500).

## Statistical analysis

All experiments were repeated at least three times; the results are presented as averages ± standard deviation (SD). Statistical significance for the difference between the datasets was calculated using one-way ANOVA with follow-up Tukey's multiple comparison tests or, when appropriate, two-way ANOVA with follow-up Sidak's multiple comparison tests. All calculations were carried out using GraphPad Prism. The p-values below 0.05 were considered statistically significant. Statistical significance level in the figures is indicated as * for $p < 0.05$, ** for $p < 0.01$, *** for $p < 0.001$, and **** for $p < 0.0001$.

## Protein expression and purification

DNA sequence encoding MEMO1 was codon optimized for *Echerichia coli* expression and prepared by chemical synthesis (Integrated DNA Technologies, Inc). Mutant variants of MEMO1 were generated by site-directed mutagenesis. MEMO1 was expressed as fusion with the chitin-binding domain and intein using vector pTYB12 (New England BioLabs) in *E. coli* BL21(DE3) and purified by chitin affinity chromatography combined with intein self-cleavage, essentially as described previously (*Dmitriev et al., 2006*). MEMO1 was additionally purified by size exclusion chromatography on a Superdex 75 10/300 GL Increase column (GE Life Sciences) in a buffer containing 50 mM HEPES-Na, pH 7.4, 150 mM NaCl, and 0.6 mM tris-(2-carboxyethyl)phosphine and concentrated by membrane filtration.

## Isothermal titration calorimetry

For ITC experiments, MEMO1 was dialyzed against 50 mM HEPES, pH 7.4, 150 mM NaCl, 9.5 mM reduced glutathione (GSH), 0.5 mM oxidized glutathione (GSSG) under argon. Iron (II) sulfate was dissolved in the used dialysis buffer. ITC was performed on a TA Instruments (New Castle, DE) Low Volume Nano calorimeter using ITCRun software and analyzed with NanoAnalyze software (TA Instruments) using independent model setting. A 0.5 mM solution of iron sulfate was titrated into 0.17 ml of 25 µM MEMO1, 1.5–2.5 µl per injection at 300 s intervals at 25°C under constant stirring at 300 r.p.m. Dilution heat correction was applied by titrating iron sulfate into the ITC buffer containing no protein. Most of the titrations were performed in duplicate or triplicate.

## Protein crystallization

Protein crystals were obtained by mixing 0.5 µl of purified MEMO1 (12–13 mg/ml) with 0.5 µl mother liquor containing 100 mM 2-($N$-morpholino) ethanesulfonic acid pH 5.5–7.0 and 22.5% PEG-3350 and incubated at 20°C. Protein crystals that contained reduced $Fe^{2+}$ and $Cu^+$ were set up inside an anaerobic chamber with 3 ppm of $O_2$ in the atmosphere of 4% $H_2/N_2$ mixture. Mother liquor containing 1 mM metal was incubated inside the chamber with mixing to get rid of oxygen dissolved in solution prior to crystallization. Crystals were harvested and stored in liquid nitrogen using 20% glycerol as cryoprotectant. Crystals diffracted to 1.75–2.55 Å and belonged to orthorhombic system, space

group P2₁2₁2, with cell dimensions of approximately a=140 Å, b=87 Å, c=98 Å, α=β=γ=90°, and contained four molecules in the asymmetric unit.

## Data collection and structure refinement

Diffraction data were collected at the Canadian Light Source, CMCF section, using beamlines 08ID and 08BM and Pilatus 6M detector. Data were integrated and scaled using XDS package to 2.15 Å (for MEMO1-Fe complex), 2.55 Å (for MEMO1-Cu complex), and 1.75 Å (for MEMO1 C244S mutant) (*Kabsch, 2010*). Initial phases were obtained using Phaser MR (*McCoy et al., 2007*) within ccp4i package and previously solved MEMO1 structure (PDB code: 3BCZ) as a model for molecular replacement (*Qiu et al., 2008*). The final models of MEMO1 were refined using phenix.refine (*Liebschner et al., 2019*) with manual rebuilding using COOT (*Emsley et al., 2010*). The coordinates and structure factors have been deposited to the Protein Data Bank with ID codes: 7KQ8 (MEMO1-Fe), 7L5C (MEMO1-Cu), and 7M8H (MEMO1 C244S).

## Materials availability statement

MEMO1 knockout and knockdown cell lines and MEMO1 expression vectors are available from the authors upon a reasonable request.

## Acknowledgements

We thank the Canadian Light Source, CMCF section staff for support during data collection. We gratefully acknowledge the use of instrumentation at the Protein Characterization, Crystallization Facility (PCCF) and Phenogenomic Imaging Centre of Saskatchewan (PICS) supported by College of Medicine, University of Saskatchewan. ICP-MS measurements were performed by Marvel Davis in the OHSU Elemental Analysis Shared Resource with partial support from NIH core grant S10RR025512. NMR spectra were recorded at the National Magnetic Resonance at Madison (NMRFAM), which is supported by NIH grant P41 GM103399 and by the University of Wisconsin-Madison. This research was supported by the Canadian Institutes of Health Research Project grant PJT-178246, Natural Sciences and Engineering Council of Canada Discovery grant RGPIN-2017-06822, and the University of Saskatchewan funding to OYD. FJV supported this work with funds from Canadian Institute of Health Research (PJT-156309), Canada Foundation for Innovation (CFI-33364) and operating grants from Saskatchewan Cancer Agency.

## Additional information

### Funding

| Funder | Grant reference number | Author |
|---|---|---|
| Canadian Institutes of Health Research | PJT-178246 | Oleg Y Dmitriev |
| Natural Sciences and Engineering Research Council of Canada | RGPIN-2017-06822 | Oleg Y Dmitriev |
| Canada Foundation for Innovation | CFI-33364 | Franco J Vizeacoumar |
| Canadian Institutes of Health Research | PJT-156309 | Franco J Vizeacoumar |
| Saskatchewan Cancer Agency | | Franco J Vizeacoumar |
| University of Saskatchewan | | Oleg Y Dmitriev |

The funders had no role in study design, data collection and interpretation, or the decision to submit the work for publication.

## Author contributions
Natalia Dolgova, Conceptualization, Data curation, Validation, Investigation, Visualization, Methodology, Writing – original draft, Writing – review and editing; Eva-Maria E Uhlemann, Conceptualization, Formal analysis, Validation, Investigation, Visualization, Methodology, Writing – original draft, Writing – review and editing; Michal T Boniecki, Resources, Data curation, Formal analysis, Validation, Investigation, Visualization, Methodology, Writing – original draft, Writing – review and editing; Frederick S Vizeacoumar, Conceptualization, Data curation, Software, Formal analysis, Validation, Investigation, Visualization, Methodology, Writing – original draft; Anjuman Ara, Paria Nouri, Syed A Abbas, Jaala Patry, Hussain Elhasasna, Investigation; Martina Ralle, Investigation, Methodology, Writing – original draft; Marco Tonelli, Investigation, Methodology; Andrew Freywald, Conceptualization, Data curation, Validation, Writing – original draft, Writing – review and editing; Franco J Vizeacoumar, Conceptualization, Resources, Data curation, Formal analysis, Supervision, Visualization, Writing – original draft, Writing – review and editing; Oleg Y Dmitriev, Conceptualization, Resources, Data curation, Formal analysis, Supervision, Funding acquisition, Validation, Investigation, Visualization, Writing – original draft, Project administration, Writing – review and editing

## Author ORCIDs
Hussain Elhasasna http://orcid.org/0000-0002-2810-3138
Franco J Vizeacoumar https://orcid.org/0000-0002-6452-5207
Oleg Y Dmitriev http://orcid.org/0000-0003-1307-5063

## Decision letter and Author response
Decision letter https://doi.org/10.7554/eLife.86354.sa1
Author response https://doi.org/10.7554/eLife.86354.sa2

# Additional files

## Supplementary files
• Supplementary file 1. Genetic interactions of MEMO1 with iron related genes. (A) Iron-related genes exhibiting gene-of-function (GOF) interactions with *MEMO1*. (B) Iron-related genes exhibiting loss-of-function (LOF) interactions with *MEMO1*. (C) Genes involved in ferroptosis and exhibiting GOF or LOF interactions (highlighted in light blue) with *MEMO1*. (D) Structure determination statistics for MEMO1-metal complexes and the C244S mutant.

• MDAR checklist

## Data availability
Diffraction data have been deposited in PDB under the accession code s 7KQ8, 7L5C, and 7M8H. All other data generated or analysed during this study are included in the manuscript and supporting file.

The following datasets were generated:

| Author(s) | Year | Dataset title | Dataset URL | Database and Identifier |
| --- | --- | --- | --- | --- |
| Boniecki MT, Uhlemann EE, Dmitriev OY | 2022 | Structure of copper bound MEMO1 | https://www.rcsb.org/structure/7L5C | RCSB Protein Data Bank, 7L5C |
| Boniecki MT, Uhlemann EE, Dmitriev OY | 2022 | Structure of iron bound MEMO1 | https://www.rcsb.org/structure/7KQ8 | RCSB Protein Data Bank, 7KQ8 |
| Boniecki MT, Uhlemann EE, Dmitriev OY | 2022 | Structure of Memo1 C244S metal binding site mutant at 1.75A | https://www.rcsb.org/structure/7M8H | RCSB Protein Data Bank, 7M8H |

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
