## [Editor Report]

This important work demonstrates a function for MEMO1, a poorly understood protein that is commonly dysregulated in cancer. They provide convincing evidence that MEMO1 binds iron, and interacts with a number of known proteins involved in iron metabolism such as transferrin and mitoferrin. It still remains unclear whether the downstream metabolic programs affected by this protein in cancer are directly related to its iron binding activity or other effects of it in cell metabolism, which should be the focus of future work.

---

## [Decision Letter]

**Decision letter after peer review:**

Thank you for submitting your article "MEMO1 is a Metal Containing Regulator of Iron Homeostasis in Cancer Cells" for consideration by *eLife*. Your article has been reviewed by 3 peer reviewers, one of whom is a member of our Board of Reviewing Editors, and the evaluation has been overseen by Richard White as the Senior Editor. The following individuals involved in the review of your submission have agreed to reveal their identity: Daniel Kosman (Reviewer #2); Thomas Benedict Bartnikas (Reviewer #3).

Essential revisions (for the authors):

1) Testing whether iron loading and chelation exacerbate or reverse effects of MEMO1 KD and KO with more than just mitochondrial morphology, e.g. cell survival, apoptosis, motility, and proliferation. This will also help to reorganize the manuscript to focus on the meaning of decreased MEMO1 in the single breast cancer model to deeply understand function rather than the presumptive effect of increased MEMO1 in TNBC.

2) A more thorough look at iron trafficking, including ferritin, PCBP1/2, BolA, and NCOA4, with an eye towards the ferroptosis connection. They also need to evaluate the expression of GPX4 and do both iron depletion and iron loading to see if the system goes in both directions when MEMO1 is KD and KO vs normal. And they can use ferroptosis blocker ferristatin-1 in these cells.

3) Clarification on the mechanistic understanding of MEMO1 function through TFR2. this is likely to require some additional focused experiments on whether there is physical interaction between them, whether TFR2 binding iron is abrogated in the presence of MEMO1, or other such additional data to shed light on the mechanism. In the absence of additional mechanistic evidence of the relationship between MEMO1 and TFR2, a more tempered set of claims/conclusions is warranted.

*Reviewer #1 (Recommendations for the authors):*

1) Does Figure 2A suggest that gene essentiality score distribution is decreased in MEMO1 high cells? What is the meaning of such a finding, that these genes become less relevant for disease when MEMO1 is elevated? Is the rest of the data consistent with this conceptually?

2) TFR2 shRNA leads to increase cell proliferation. The authors state that is "possibly due to the increased availability of iron." This does not appear to be well substantiated by the data or the to-date known function of TFR2. Please edit/clarify/remove.

3) Loss of MEMO1 and TFR2 leads to decreased proliferation in both breast and melanoma cell lines. Unclear how specific the meaning is in such a situation. Please clarify as this reviewer anticipated that if MEMO1 is relevant only in breast cancer and melanoma serves as a negative control, how is the cooperation between MEMO1 and TFR2 present in melanoma cells?

4) Correlation of 0.2 remains weak and the difference between breast and melanoma cell lines is not evident. The P values provide evidence that the confidence in the weak correlation is strong but do not provide evidence of a strong correlation. This is at best misleading. Please remove.

5) Please provide gels for Figure 3C-3F. Unclear how Figure 3 is different from Figure 2; is it the same Western blots?

6) If MEMO1 knockdown leads to decreased expression of most genes analyzed in the iron regulatory pathways and decreased expression of these genes also decreases MEMO1 expression (more specifically in the breast relative to the melanoma cell lines), what is the hypothesis about their relatedness? Also, PLOD1 is difficult to understand; looks significantly different but in opposite directions for KD vs KO. What is a possible interpretation?

7) While the authors use a single breast and compare it to a single melanoma cell line, is it possible that these changes are specific to a single cell line and do not represent breast cancer as a whole? The sentence on page 11, line 255 is thus an overstatement.

8) Figure 4 shows decreased cytosolic and mitochondrial iron in MEMO1 knockout cells. We already see in Figure 3 that MEMO1 knockdown results in the downregulation of TFR1 which is the likely cause of decreased subcellular iron. Additionally, to conjecture that this means that MEMO1 overexpression must be involved in iron accumulation is an overstatement, and to postulate that it is TFR2 (rather than TFR1 mediated) when TFR1 rather than TFR2 is involved in iron uptake into cells is incorrect.

9) The finding in Figure 4C is highly intriguing. What do the authors think this could mean physiologically? Why would a presumed iron mediating protein bind apoTF with higher affinity than holoTF when apoTF is devoid of iron? Can the authors induce mutation in MEMO1 that abrogates binding to apoTF to test whether this is an essential component of MEMO1 functioning?

10) Expression of ferritin (mRNA and protein) would assist in corroborating how much iron is stored in these cells and expression of iron chaperones (PCBP1/2 and NCOA4) is also important to confirm that changes in iron imported via TFR1 is consistent with changes in ferritinophagy by NCOA4. These are critical to understanding iron in the context of ferroptosis.

11) The authors refer to the "labile iron pool" which does not exist in physiological conditions intracellularly and the conjecture that DFX depletes it (page 13, line 297) is misguided. For example, DFX has a higher affinity for iron relative to transferrin and would be expected to outcompete it to bind iron; that said, TF-bound iron is not at all labile, and thus intracellular iron does not need to be labile to be available for binding DFX. Looking at other iron chelators would be helpful as DFP for example is able to chelate intracellular iron (while DFX is presumed to bind iron extracellularly). Finally, iron poor conditions, by modulating TF saturation in the culture media, would be another way of depleting iron in these cells. This is central to the hypothesis of the entire manuscript and requires a more thorough approach.

12) Figure 4E suggests that the difference is significant only between the black and red lines but no statistics are presented. Please add stats and a number of independent runs of this experiment. Furthermore, what is the Y-axis? In other words, what is fluorescence measuring? Presumably apoptosis or cell death? And only M67-9, not M67-2 cells were used; can M67-2 cells also be shown here as the expectation is that they will demonstrate an even bigger difference from MDA cells, right? Finally, was GPX4 expression or activity measured in MDA vs M67-9 vs M67-2 cells or A-375 vs A67-16 cells? Please add.

13) Figure 4F is very confusing. Measuring MDA as a surrogate of lipid peroxidation demonstrates that a decrease and loss of MEMO1 lead to an increase in lipid peroxidation. To reconcile with Figure 4E, a potential hypothesis would be that more lipid peroxidation DOES NOT lead to increased cell death. Please provide evidence that cell death is in fact decreased. It would also be more clear to add iron back to see if the effect of MEMO1 KD or KO is lost when there is increased iron in the medium or by overexpressing TFR1 for example. Finally, how do the authors reconcile that only A67-4 but not A67-16 has an effect on lipid peroxidation relative to A-375 cells and M67-9 may have a lesser effect on lipid peroxidation than M67-2 with the hypothesis that the effect is MEMO1 driven?

14) While the authors knockdown or knockout MEMO1, their main focus is on MEMO1 increased expression. This is not a trivial difference as their own data shows that sometimes even knockdown has a different direction of effect relative to knockdown and in physiology, finding that decreasing something has effect X does not directly lead to the assumption that increasing it will lead to the opposite of X. This is a central conceptual weakness of the manuscript.

*Reviewer #2 (Recommendations for the authors):*

This work provides a thorough 'first look' at a protein that likely plays a previously unknown role in cell iron homeostasis. The 'look' is experimentally broad, from gene interaction analysis to structural and biophysical analysis in support of this conclusion. There are points that I think bear consideration by the authors but, experimentally, the work is first-rate.

1. While the hypothesis that MEMO1 plays a key role in cell iron homeostasis remains to be directly tested, the data presented herein clearly support further delineation of the underlying mechanisms. The key findings in this regard are the facts, as established herein, that:(1) MEMO1 binds ferrous iron (the appropriate valence state for cell iron) along with glutathione (Figure 5A); (2) the structure of MEMO1 in complex with Fe(II)-GSH reveals the coordination site within the protein for this complex (Figure 5B/c); (3) oxidative stress and sensitivity to ferroptosis correlate with MEMO1 protein abundance in a consistent fashion (Figure 4); and (4) while the effect is limited, there are data that indicate a relation between cell iron content and MEMO1 abundance (Figure 4A/B).

2. I find the data presented in Figure 3 less compelling with respect to a link between MEMO1 abundance and the transcript abundance of various iron-related transcripts. The differences are small, and as far as I can deduce from the very small typeface, of limited statistical significance overall. In contrast, the knock-down of these iron-related transcripts by shRNA in breast cancer cells had a consistent, statistically significant effect on MEMO1 expression. Together with the gene interaction analysis that got this work started, the data do support a model wherein MEMO1 modulates iron metabolism in some fashion, at the least.

3. A key experiment is cytoplasmic and mitochondrial iron content as a function of MEMO1 (Figure 4A/B). While, as noted above, there is a correlation in that as MEMO1 abundance decreases, iron content decreases. A question I have about the cytoplasmic values is that they reflect total iron and not 'labile' iron, eg, the values would include ferritin iron. And, along with the ferritin, I question the contribution of lysosomal iron to that total. The authors refer to their data in 4A as the 'labile iron pool.' Their ICP-MS approach does not interrogate this pool specifically. There are Fe(II) specific fluorescent dyes that do so and such data would be appropriate here.

4. This brings me to the point that the authors need to spend more time in thinking about MEMO1 and cell iron metabolism. They make no mention at all of the PCBP1/2, GSH, BolA cytoplasmic iron trafficking machinery. Indeed, the binding of Fe(II)-GSH to MEMO1 looks a lot like the binding of Fe(II)-GSH to the PCBP1-BalA complex. Furthermore, the Fe(II) dissociation constants are quantitatively similar as well. In short, IF MEMO1 is coordinating Fe(II)-GSH in the cytoplasm as postulated, it is doing so either in competition with or in conjunction with the PCBP1-BolA pathway to the assembly of the cytoplasmic Fe,S clusters, including the one that regulates ACO1/IREBP1.

5. Using microscale thermophoresis, the authors quantify a fairly robust interaction between MEMO1 and apo- and holo-transferrin (Tf). This, together with the genetic interaction between MEMO1 and TFR is taken in support of the premise that MEMO1 plays "an important role for MEMO1 in maintaining iron homeostasis." I would expect the authors to explain in further detail how a MEMO1 interaction with Tf occurs if MEMO1 is not said to be in a vesicular compartment, not linked in any way to the endolysosomal trafficking pathway associated with Tf-TFR iron processing. The authors also use "interaction" indiscriminately, without distinguishing between genetic interaction, which is 'in silico' and physical, which is biophysically quantified.

6. The authors refer to the 'kiss and run' hypothesis (albeit referring to it as 'novel' although it's been around for two decades). Recent work from the Barroso lab makes a good case that DMT1 provides a bridge that delivers endolysosomal iron directly to mitochondria. The authors might consider how MEMO1 might interact with that mechanism.

Experimentally, it is thorough and well-documented and offers a new look at a protein that has been at the edges of iron metabolism (and copper, but I agree with the authors that this is not likely to be the case). This work and its subject will stimulate much further research which is just what an article in a journal like *eLife* is meant to do.

*Reviewer #3 (Recommendations for the authors):*

I have included below a list of comments that I believe, when addressed, will strengthen the manuscript.

– On lines 129-131, the manuscript states "This analysis yielded genes that become highly essential (Wilcoxon-rank sum test P<0.05) when MEMO1 is either under-expressed, representing GIs identified from loss of function (LOF-GIs), or overexpressed, representing GIs identified from a gain of function, (GOF-GIs).". Is this gene list included in the paper? If not, please include it

– In Figure 2, * is shown in multiple places. I assume this indicates a significant difference, but this isn't described in the legend.

– On lines 201-22, the manuscript states "TFR2, but not TFR1, was found to exhibit GI with MEMO1 along with other iron-related genes (Figure 2A)." TFR1 data are not shown in Figure 2A.

– Figure 3A and B include a dashed line which the text mentions indicate an arbitrary cut-off of R=0.2. Please add this description to the legend as well

– In the Results section entitled "MEMO1 plays an important role in the maintenance of iron concentration in mitochondria", the authors describe that they analyzed the physical interaction between MEMO1 and transferrin. It's not clear to this reviewer why an interaction between MEMO1 and transferrin was assessed, but not an interaction between MEMO1 and TFR2 given that MEMO1 and TFR2 were shown to interact genetically. Also, the observation that MEMO1 has a greater affinity for apo- than holo-transferrin is curious, and should be addressed in the discussion. On a related note, the statement on lines 502-503 that "transferrin and TFR2… showed physical or genetic interactions with MEMO1 in our work" needs to be clarified. Finally, the fact that MEMO1 is a cytosolic protein and transferrin resides either in the extracellular space or in the endosomal pathway suggests that a MEMO1-transferrin interaction is not physiologically relevant-- please comment.

– The authors measure iron levels in cell lines, but MEMO1 also has published roles in copper biology-- copper levels need to be measured in this report as well.

– The observation that MEMO1 knockout in cancer cells increases ferroptosis resistance AND lipid oxidation AND decreases iron levels is intriguing to this reviewer. If less iron is present in the cell, why would there be greater lipid oxidation, if lipid oxidation is dependent upon iron? Also, do the authors know how the cells are more tolerant to greater lipid oxidation? These last question needs to be addressed in the Discussion section.

– On lines 400-401, the authors state "Thus, metal coordination in MEMO1 is achieved by H49, H81, and C244, while H192 and D189 do not participate in metal binding in MEMO1". However, in Figure 5A, the ITC curves for H192A and wild-type differ quite a bit-- wouldn't this suggest that H192A has some sort of impact?

Please include statistical analysis for Figure S5-S9.

[Editors' note: further revisions were suggested prior to acceptance, as described below.]

Thank you for resubmitting the paper entitled "MEMO1 is a Metal Containing Regulator of Iron Homeostasis in Cancer Cells" for further consideration by *eLife*. Your revised article has been evaluated by a Senior Editor and a Reviewing Editor. We are sorry to say that we have decided that this submission will not be considered further for publication by Life.

Dolgova et al. present a well-written manuscript focused on the mechanism of MEMO1 function in tumor cells. The authors explore whether the mechanism of MEMO1 overexpression in breast cancer, especially TNBC, are related to regulating iron given evidence that MEMO1 binds multiple proteins in the iron regulation pathway. While the data is in part compelling, the claims are based on preliminary evidence using gene linkage and ITC to assert an iron-mediated central role of MEMO1 in tumorogenesis and perhaps metastasis and a mechanism is yet clear. Additional data was provided in the revised manuscript but increases confusion without a substantive contribution to the essential revisions which were at best partially addressed and data was removed from the manuscript, making the current manuscript version incomplete to warrant a version of record.

*Reviewer #1 (Recommendations for the authors):*

Dolgova et al. present a well-written manuscript focused on the mechanism of MEMO1 function in tumor cells. The authors explore whether the mechanism of MEMO1 overexpression in breast cancer, especially TNBC, is related to regulating iron given evidence that MEMO1 binds multiple proteins in the iron regulation pathway. While the data is in part compelling, the claims are based on indirect evidence for a central role of MEMO1 in tumorogenesis and perhaps metastasis via its effect on iron homeostasis.

The authors have attempted to address the comments raised by the reviewers. However, in some cases, their technical difficulties prevented clarification and in others, there was no additional data added to confirm/refute the idea, leading the authors to remove some of the data presented in the original submission. As a consequence, the manuscript is left weaker rather than stronger. Specifically, the transferrin data has been removed and the conceptual relationship with TFR2 is tempered. The argument that MEMO1 allows cells to accumulate more iron is inconsistent with elevated TFR1 as TFR1 is typically decreased in iron loaded cells. The reason TFR1 would be upregulated as a mechanism for iron accumulation is not made clear. It is also not clear what the meaning is of Figure 4C; while the authors describe superficially that high MEMO cells are more sensitive to ferroptosis, the figure and adjoining text do not make this clear. Taken together, the manuscript is premature and more superficial than previously. While there is some interesting and compelling preliminary data, there is insufficient evidence for a complete story and may be better suited in a cancer journal for a more targeted audience.

*Reviewer #3 (Recommendations for the authors):*

The authors did address essential revision #1. They attempted to address essential revision #2 but there were 'technical difficulties' of an unspecified nature. The authors addressed essential revision #3 by tempering their claims of a TFR2-MEMO1 interaction.

With regards to my specific recommendations, they addressed most but not all. First, I did ask for asterisks to be defined in the legends, as this is missing from several. They responded that the symbols are defined in the methods, but this makes the paper less accessible to the reader. Adding a one-sentence description to a legend will not compromise the brevity of the manuscript. Second, I asked for statistical analysis to be performed on supplemental figures, and that request was denied.

Overall, the authors did address some of the issues raised in the initial review. However, the new data that were added do not help clarify the role of MEMO1 in cellular iron homeostasis.

[Editors' note: further revisions were suggested prior to acceptance, as described below.]

Thank you for resubmitting your work entitled "MEMO1 is a Metal Containing Regulator of Iron Homeostasis in Cancer Cells" for further consideration by *eLife*. Your revised article has been evaluated by Richard White (Senior Editor) and a Reviewing Editor.

Overall, the reviewers agree that the has been improved. As you can see below, both reviewers raise the important point of whether MEMO1 is really a direct regulator of iron homeostasis. Based on their suggestions, we would request that the title be changed to better show what is mechanistically known from your studies. Please have a look at the below comments and propose an alternative title that reflects their comments.

*Reviewer #2 (Recommendations for the authors):*

Having reviewed this manuscript twice previously, my comments concerning this submission can be brief. In the Abstract, the authors state: "We conclude that MEMO1 is an iron-binding protein that regulates iron homeostasis in cancer cells." The data presented directly and substantially support that conclusion. The data presented are novel in that they are the first to provide any evidence for the function of this protein while definitively providing data on the protein's structure and metal binding properties. Genetically, the protein is linked to a variety of proteins critical to cell iron metabolism, and the protein is linked to "iron behaving badly" in cells – ferroptosis- although the work does not elucidate what that connection is except for the inference that cell content of MEMO1 correlates with the abundance of cell iron and that represents the ferroptotic link. This is inference is plausible albeit not experimentally confirmed by the work presented. Given that the authors are clear that the cellular observations in this work are restricted to "cancer cells" it is inappropriate to criticize the work as not reflecting a possible MEMO1 function in your garden variety enterocyte or hepatocyte. I support publication of this manuscript as is in *eLife*.

*Reviewer #3 (Recommendations for the authors):*

The authors have addressed my comments from the last round of review. However, I view this manuscript only as a first step in understanding the potential role of MEMO1 in iron metabolism in cancer cells, given that the results of this study do not establish the specific function of MEMO1 in iron metabolism. There is the distinct possibility that MEMO1 doesn't play a direct role in iron metabolism, and that the phenotypes uncovered in this study reflect far downstream consequences of altered MEMO1 expression. Along those lines, if the editors do accept this paper for publication, I would make one last suggestion-- revise the title of the paper. Without knowing the specific function of MEMO1, it is too early to declare that it regulates iron homeostasis.

---

## [Author Response]

Essential revisions (for the authors):1) Testing whether iron loading and chelation exacerbate or reverse effects of MEMO1 KD and KO with more than just mitochondrial morphology, e.g. cell survival, apoptosis, motility, and proliferation. This will also help to reorganize the manuscript to focus on the meaning of decreased MEMO1 in the single breast cancer model to deeply understand function rather than the presumptive effect of increased MEMO1 in TNBC.

As requested by the reviewers, we have tested the effect of the cell permeable iron chelator deferiprone on the viability of the parental cells and MEMO1 knockouts and found the high-MEMO1 cells to be slightly more sensitive to deferiprone. We have then tested a high subtoxic concentration of deferiprone (30 μM) on cell proliferation and cell motility in breast cancer cell lines and did not find any significant effect in either the parental cell line or MEMO1 knockout or knockdown (Figure 4 —figure supplement 2). This is perhaps not surprising, because iron binding affinity of deferiprone is much higher than that of MEMO1. So, acting inside the cell, deferiprone will deplete both the labile iron pool and MEMO1 before any high-affinity iron-binding proteins, in effect equalizing high-MEMO1 and no-MEMO1 cells. On the other hand, high-MEMO1 cells turned out to be much more sensitive to iron overload than MEMO1 knockout, and this effect appears to be linked to ferroptosis. This data has been included in the revised manuscript (Figure 4C).

2) A more thorough look at iron trafficking, including ferritin, PCBP1/2, BolA, and NCOA4, with an eye towards the ferroptosis connection. They also need to evaluate the expression of GPX4 and do both iron depletion and iron loading to see if the system goes in both directions when MEMO1 is KD and KO vs normal. And they can use ferroptosis blocker ferristatin-1 in these cells.

PCBP1 and PCBP2 did not show any genetic interactions with MEMO1 in our genome wide database screening of gene essentiality. On the other hand, NCOA4 was among the loss-of-function hits, while BOLA2 was among the gain-of-function hits. This data has been included in the manuscript. Unfortunately, our attempts to validate the screening results experimentally met with technical difficulties, and to date we have been unable to verifiably knock down expression of BOLA2 and NCOA4 in our cell lines. This work will continue as a part of our ongoing investigation of MEMO1 molecular function. We have included the data on GPX4 and ferritin expression, as well as ferrostatin-1 effect on cell viability at various iron levels in the revised manuscript (Figure 4C and Figure 4 – supplement 1). This data strengthens our conclusion that MEMO1 is involved in ferroptosis and suggests that the higher sensitivity of high-MEMO1 cells to ferroptosis is not due to lower GPX4 expression, but rather to the altered levels and distribution of iron in the cell.

3) Clarification on the mechanistic understanding of MEMO1 function through TFR2. this is likely to require some additional focused experiments on whether there is physical interaction between them, whether TFR2 binding iron is abrogated in the presence of MEMO1, or other such additional data to shed light on the mechanism. In the absence of additional mechanistic evidence of the relationship between MEMO1 and TFR2, a more tempered set of claims/conclusions is warranted.

The mechanism of TFR2 -MEMO1 interaction is currently under investigation. Following the reviewers’ suggestions, we tempered our conclusions in this respect.

Reviewer #1 (Recommendations for the authors):1) Does Figure 2A suggest that gene essentiality score distribution is decreased in MEMO1 high cells? What is the meaning of such a finding, that these genes become less relevant for disease when MEMO1 is elevated? Is the rest of the data consistent with this conceptually?

A negative essentiality score means that the cells with the gene knockdown or knockout lose viability or proliferate slower than the control. This has been clarified in the figure2 legend. So, more negative scores in the high-MEMO1 cell lines compared to the low-MEMO1 cell lines mean that the genes are more essential in the former. Thus, these results are consistent with the rest of the data.

2) TFR2 shRNA leads to increase cell proliferation. The authors state that is "possibly due to the increased availability of iron." This does not appear to be well substantiated by the data or the to-date known function of TFR2. Please edit/clarify/remove.

This part of the sentence has been removed.

3) Loss of MEMO1 and TFR2 leads to decreased proliferation in both breast and melanoma cell lines. Unclear how specific the meaning is in such a situation. Please clarify as this reviewer anticipated that if MEMO1 is relevant only in breast cancer and melanoma serves as a negative control, how is the cooperation between MEMO1 and TFR2 present in melanoma cells?

We do not think that MEMO1 function is only important in breast cancer cells, rather that breast cancer cells may be more sensitive to the disruption of iron homeostasis, and therefore the effects of MEMO1 knockout or knockdown are more pronounced than in melanoma. In fact, the difference in the proliferation rates +/- Tfr2 between the high-MEMO1 cells and MEMO1 knockout or knockdown is smaller in melanoma than in breast cancer cells (Figure 2C).

4) Correlation of 0.2 remains weak and the difference between breast and melanoma cell lines is not evident. The P values provide evidence that the confidence in the weak correlation is strong but do not provide evidence of a strong correlation. This is at best misleading. Please remove.

We agree that the correlation is weak, but this simply reflects the fact that our analysis did not attempt to identify a distinct subtype of breast cancer cell lines that shows a strong correlation between MEMO1 and the second gene expression levels. Based on the high confidence in the correlation in breast cancer provided by the P-values, we prefer to keep this data in the manuscript.

5) Please provide gels for Figure 3C-3F. Unclear how Figure 3 is different from Figure 2; is it the same Western blots?

Figure 2 shows representative Western blots as evidence of effective gene expression suppression by the respective shRNAs. Figure 3 shows the results of the quantitative analysis of at least three different blots (biological replicates), including the ones shown in Figure 2. This is clarified in the legend to Figure 3

6) If MEMO1 knockdown leads to decreased expression of most genes analyzed in the iron regulatory pathways and decreased expression of these genes also decreases MEMO1 expression (more specifically in the breast relative to the melanoma cell lines), what is the hypothesis about their relatedness? Also, PLOD1 is difficult to understand; looks significantly different but in opposite directions for KD vs KO. What is a possible interpretation?

The observed correlation suggests co-regulation of these genes and supports a hub role for MEMO1 in iron homeostasis, as discussed in the manuscript, but the mechanism awaits further investigation. Since we don’t know the mechanism of MEMO1 dependent regulation yet, we prefer not to speculate about the PLOD1 expression level changes.

7) While the authors use a single breast and compare it to a single melanoma cell line, is it possible that these changes are specific to a single cell line and do not represent breast cancer as a whole? The sentence on page 11, line 255 is thus an overstatement.

We agree that this an overstatement and this sentence has been changed accordingly.

8) Figure 4 shows decreased cytosolic and mitochondrial iron in MEMO1 knockout cells. We already see in Figure 3 that MEMO1 knockdown results in the downregulation of TFR1 which is the likely cause of decreased subcellular iron. Additionally, to conjecture that this means that MEMO1 overexpression must be involved in iron accumulation is an overstatement, and to postulate that it is TFR2 (rather than TFR1 mediated) when TFR1 rather than TFR2 is involved in iron uptake into cells is incorrect.

Indeed, TFR2 may not directly drive bulk iron uptake, but based on its genetic interaction with MEMO1, a regulatory TFR2-MEMO1 pathway seems to be plausible, even if the decrease in cellular iron is ultimately due to a decreased uptake through TFR1. The respective passage has been clarified.

9) The finding in Figure 4C is highly intriguing. What do the authors think this could mean physiologically? Why would a presumed iron mediating protein bind apoTF with higher affinity than holoTF when apoTF is devoid of iron? Can the authors induce mutation in MEMO1 that abrogates binding to apoTF to test whether this is an essential component of MEMO1 functioning?

We appreciate this suggestion. This is an excellent way to address the question about the physiological relevance of the MEMO1-TF interaction. This work is in progress. In fact, in view of the reviewers’ comments, we feel that the interaction between MEMO1 and transferrin requires additional investigation in the cell contex tand decided not to include the in vitro data in the revised manuscript.

10) Expression of ferritin (mRNA and protein) would assist in corroborating how much iron is stored in these cells and expression of iron chaperones (PCBP1/2 and NCOA4) is also important to confirm that changes in iron imported via TFR1 is consistent with changes in ferritinophagy by NCOA4. These are critical to understanding iron in the context of ferroptosis.

These are all excellent suggestions that we will follow in our ongoing efforts to understand the mechanism of iron homeostasis regulation by MEMO1. For now, we have included ferritin expression data (protein levels) in Figure 4 -supplement.

11) The authors refer to the "labile iron pool" which does not exist in physiological conditions intracellularly and the conjecture that DFX depletes it (page 13, line 297) is misguided. For example, DFX has a higher affinity for iron relative to transferrin and would be expected to outcompete it to bind iron; that said, TF-bound iron is not at all labile, and thus intracellular iron does not need to be labile to be available for binding DFX. Looking at other iron chelators would be helpful as DFP for example is able to chelate intracellular iron (while DFX is presumed to bind iron extracellularly). Finally, iron poor conditions, by modulating TF saturation in the culture media, would be another way of depleting iron in these cells. This is central to the hypothesis of the entire manuscript and requires a more thorough approach.

Our logic in using deferoxamine was to create iron limiting conditions in the medium, since DFX is indeed presumed to bind iron outside the cell. This seems to be a more physiologically relevant approach to stressing iron homeostasis in the cell than using high concentrations of the cell-permeable iron chelators. However, we agree that testing a cell permeable chelator can be informative too, and we have conducted the experiments with deferiprone, as suggested by the reviewer. This data is shown in Figure 4 – supplement 2.

As a clarification, we do not use the term labile iron pool as a synonym for free intracellular iron, but rather for exchangeable iron bound to low-to-moderate affinity acceptors, such as glutathione (Kd about 10-5 M), or iron chaperone PCBP1 (Kd about 10-6 M). Presumably, this pool will be depleted first under the iron limiting conditions.

12) Figure 4E suggests that the difference is significant only between the black and red lines but no statistics are presented. Please add stats and a number of independent runs of this experiment. Furthermore, what is the Y-axis? In other words, what is fluorescence measuring? Presumably apoptosis or cell death? And only M67-9, not M67-2 cells were used; can M67-2 cells also be shown here as the expectation is that they will demonstrate an even bigger difference from MDA cells, right? Finally, was GPX4 expression or activity measured in MDA vs M67-9 vs M67-2 cells or A-375 vs A67-16 cells? Please add.

As requested, we have added statistics (Figure 4E) and GPX4 expression data (Figure 4 – supplement 1). We clarified in the Figure 4 legend that cell viability was measured by resazurin fluorescence. M67-2 is the MEMO1 knockdown, and M67-9 is the knockout, so the largest difference is expected to be between the MDA-MB-231 and M67-9, as shown. We have amended the graph legend to avoid confusion.

13) Figure 4F is very confusing. Measuring MDA as a surrogate of lipid peroxidation demonstrates that a decrease and loss of MEMO1 lead to an increase in lipid peroxidation. To reconcile with Figure 4E, a potential hypothesis would be that more lipid peroxidation DOES NOT lead to increased cell death. Please provide evidence that cell death is in fact decreased. It would also be more clear to add iron back to see if the effect of MEMO1 KD or KO is lost when there is increased iron in the medium or by overexpressing TFR1 for example. Finally, how do the authors reconcile that only A67-4 but not A67-16 has an effect on lipid peroxidation relative to A-375 cells and M67-9 may have a lesser effect on lipid peroxidation than M67-2 with the hypothesis that the effect is MEMO1 driven?

The MDA levels were measured in the cells grown in the standard medium without RSL3, iron, or iron chelators added. Under these conditions, M67-9 (KO) and M67-2 (KD) cell proliferate slightly slower than MDA-MB-231 (WT) (Figure 1- supplement 5), but otherwise normally (Figure S5D). This means that even though MDA and therefore lipid peroxides are elevated in the KO and KD cells, they do not reach highly toxic levels that would cause cell death. The difference in lipid peroxidation levels between the knockouts and knockdowns is quite small and may result from a combination of several secondary opposite effects caused by the changes in MEMO1 level.

14) While the authors knockdown or knockout MEMO1, their main focus is on MEMO1 increased expression. This is not a trivial difference as their own data shows that sometimes even knockdown has a different direction of effect relative to knockdown and in physiology, finding that decreasing something has effect X does not directly lead to the assumption that increasing it will lead to the opposite of X. This is a central conceptual weakness of the manuscript.

We understand the difference, and our focus is not so much on MEMO1 overexpression, as on the differential effects between high and low MEMO1 expression levels in the cell as an approach to understanding MEMO1 function. Our database screening of MEMO1 genetic interactions, which served as a foundation for subsequent experimental work, in particular is free from the “knockout bias” because we compare groups of multiple cell lines that naturally differ in MEMO1 expression levels.

Reviewer #2 (Recommendations for the authors):This work provides a thorough 'first look' at a protein that likely plays a previously unknown role in cell iron homeostasis. The 'look' is experimentally broad, from gene interaction analysis to structural and biophysical analysis in support of this conclusion. There are points that I think bear consideration by the authors but, experimentally, the work is first-rate.1. While the hypothesis that MEMO1 plays a key role in cell iron homeostasis remains to be directly tested, the data presented herein clearly support further delineation of the underlying mechanisms. The key findings in this regard are the facts, as established herein, that:(1) MEMO1 binds ferrous iron (the appropriate valence state for cell iron) along with glutathione (Figure 5A); (2) the structure of MEMO1 in complex with Fe(II)-GSH reveals the coordination site within the protein for this complex (Figure 5B/c); (3) oxidative stress and sensitivity to ferroptosis correlate with MEMO1 protein abundance in a consistent fashion (Figure 4); and (4) while the effect is limited, there are data that indicate a relation between cell iron content and MEMO1 abundance (Figure 4A/B).2. I find the data presented in Figure 3 less compelling with respect to a link between MEMO1 abundance and the transcript abundance of various iron-related transcripts. The differences are small, and as far as I can deduce from the very small typeface, of limited statistical significance overall. In contrast, the knock-down of these iron-related transcripts by shRNA in breast cancer cells had a consistent, statistically significant effect on MEMO1 expression. Together with the gene interaction analysis that got this work started, the data do support a model wherein MEMO1 modulates iron metabolism in some fashion, at the least.

We apologize for a possible confusion caused by the imprecise figure labels in the original manuscript. The data shown in Figure 3 C-F are actually protein levels quantified by Western blots. We agree that the effects are small, but we think they are worth reporting since the question about a possible co-regulation of the expression of MEMO1 and its genetic interaction partners will inevitably be asked.

3. A key experiment is cytoplasmic and mitochondrial iron content as a function of MEMO1 (Figure 4A/B). While, as noted above, there is a correlation in that as MEMO1 abundance decreases, iron content decreases. A question I have about the cytoplasmic values is that they reflect total iron and not 'labile' iron, eg, the values would include ferritin iron. And, along with the ferritin, I question the contribution of lysosomal iron to that total. The authors refer to their data in 4A as the 'labile iron pool.' Their ICP-MS approach does not interrogate this pool specifically. There are Fe(II) specific fluorescent dyes that do so and such data would be appropriate here.

This is correct. We only distinguish between the cytosolic and mitochondrial fractions in our iron measurements by ICP-MS, and the cytosolic fraction includes ferritin, which is not a part of the labile iron pool. We do not equate cytosolic iron with labile iron. This has been clarified. We appreciate the suggestion to use iron-specific fluorescent dyes to test the labile iron pool.

4. This brings me to the point that the authors need to spend more time in thinking about MEMO1 and cell iron metabolism. They make no mention at all of the PCBP1/2, GSH, BolA cytoplasmic iron trafficking machinery. Indeed, the binding of Fe(II)-GSH to MEMO1 looks a lot like the binding of Fe(II)-GSH to the PCBP1-BalA complex. Furthermore, the Fe(II) dissociation constants are quantitatively similar as well. In short, IF MEMO1 is coordinating Fe(II)-GSH in the cytoplasm as postulated, it is doing so either in competition with or in conjunction with the PCBP1-BolA pathway to the assembly of the cytoplasmic Fe,S clusters, including the one that regulates ACO1/IREBP1.

This is indeed a logical suggestion. Our genome wide database screening did not reveal any genetic interactions between MEMO1 and either PCBP1 or PCBP2, but we did find a GOF interaction between MEMO1 and BOLA2, and a LOF interaction between MEMO1 and NCOA4. We are investigating the putative link between MEMO1 and these two proteins, but we have encountered technical difficulties in knocking down these genes in our cell lines and in verifying the knockdowns. Since the connection between MEMO1 and BOLA2 and NCOA4 bears more on the mechanism of iron homeostasis regulation by MEMO1 than on our central conclusion that MEMO1 is involved in such regulation, we decided to proceed with resubmission before these experiments can be completed.

5. Using microscale thermophoresis, the authors quantify a fairly robust interaction between MEMO1 and apo- and holo-transferrin (Tf). This, together with the genetic interaction between MEMO1 and TFR is taken in support of the premise that MEMO1 plays "an important role for MEMO1 in maintaining iron homeostasis." I would expect the authors to explain in further detail how a MEMO1 interaction with Tf occurs if MEMO1 is not said to be in a vesicular compartment, not linked in any way to the endolysosomal trafficking pathway associated with Tf-TFR iron processing. The authors also use "interaction" indiscriminately, without distinguishing between genetic interaction, which is 'in silico' and physical, which is biophysically quantified.

This is certainly a valid point. We agree that this data provides more questions than answers. Physiological relevance of the TF-MEMO1 interaction observed in vitro remains to be investigated, and this work is now in progress. In view of the reviewers’ comments and to improve the logic of the narrative, we decided not to include the in vitro data on MEMO1 interaction with transferrin in the revised manuscript.

6. The authors refer to the 'kiss and run' hypothesis (albeit referring to it as 'novel' although it's been around for two decades). Recent work from the Barroso lab makes a good case that DMT1 provides a bridge that delivers endolysosomal iron directly to mitochondria. The authors might consider how MEMO1 might interact with that mechanism.Experimentally, it is thorough and well-documented and offers a new look at a protein that has been at the edges of iron metabolism (and copper, but I agree with the authors that this is not likely to be the case). This work and its subject will stimulate much further research which is just what an article in a journal like eLife is meant to do.

This is certainly an interesting possibility to investigate in the future. DMT1 does show a weak but statistically significant genetic interaction with MEMO1.

Reviewer #3 (Recommendations for the authors):I have included below a list of comments that I believe, when addressed, will strengthen the manuscript.– On lines 129-131, the manuscript states "This analysis yielded genes that become highly essential (Wilcoxon-rank sum test P<0.05) when MEMO1 is either under-expressed, representing GIs identified from loss of function (LOF-GIs), or overexpressed, representing GIs identified from a gain of function, (GOF-GIs).". Is this gene list included in the paper? If not, please include it

We have included the list of iron related genes that show interactions with MEMO1. We prefer not to include the full list in this manuscript, as we continue to analyze it for a future publication.

– In Figure 2, * is shown in multiple places. I assume this indicates a significant difference, but this isn't described in the legend.

This is done for brevity. The symbols are defined in the Statistical Analysis section of the Methods.

– On lines 201-22, the manuscript states "TFR2, but not TFR1, was found to exhibit GI with MEMO1 along with other iron-related genes (Figure 2A)." TFR1 data are not shown in Figure 2A.

This is correct. We only have this type of data for the screening hits, i.e., genes that show a statistically significant difference (P<0.05) in the essentiality scores between the high- and low-MEMO1 cell lines. TFR1 is absolutely essential for cell survival regardless of MEMO1 expression level (Figure 2 -supplement 1), and therefore it was not detected in the database screen of the genetic interactions.

– Figure 3A and B include a dashed line which the text mentions indicate an arbitrary cut-off of R=0.2. Please add this description to the legend as well

The line is more visible in the high-resolution version of the figure included with the revised manuscript. The description has been added to the legend.

– In the Results section entitled "MEMO1 plays an important role in the maintenance of iron concentration in mitochondria", the authors describe that they analyzed the physical interaction between MEMO1 and transferrin. It's not clear to this reviewer why an interaction between MEMO1 and transferrin was assessed, but not an interaction between MEMO1 and TFR2 given that MEMO1 and TFR2 were shown to interact genetically. Also, the observation that MEMO1 has a greater affinity for apo- than holo-transferrin is curious, and should be addressed in the discussion. On a related note, the statement on lines 502-503 that "transferrin and TFR2… showed physical or genetic interactions withMEMO1 in our work" needs to be clarified. Finally, the fact that MEMO1 is a cytosolic protein and transferrin resides either in the extracellular space or in the endosomal pathway suggests that a MEMO1-transferrin interaction is not physiologically relevant-- please comment.

As discussed above, we fully agree that the interaction between MEMO1 and transferrin needs to be investigated in the cell context, to understand the relevance of our in vitro data. We decided not to include this data in the revised manuscript. We intend to investigate a physical interaction between MEMO1 and TFR2 in the immediate future.

– The authors measure iron levels in cell lines, but MEMO1 also has published roles in copper biology-- copper levels need to be measured in this report as well.

As a part of our total effort to understand the role of MEMO1 in metal biology, this is certainly relevant, but the focus of this manuscript is on the iron connection, and the copper measurements would lack proper context. We feel that the link between MEMO1 and copper levels in the cell requires a thorough investigation that is beyond the scope of this study.

– The observation that MEMO1 knockout in cancer cells increases ferroptosis resistance AND lipid oxidation AND decreases iron levels is intriguing to this reviewer. If less iron is present in the cell, why would there be greater lipid oxidation, if lipid oxidation is dependent upon iron? Also, do the authors know how the cells are more tolerant to greater lipid oxidation? These last question needs to be addressed in the Discussion section.

This is indeed a counter-intuitive result. One possible answer is that MEMO1-dependent iron distribution in the cell may affect lipid oxidation levels more than the total iron level, as discussed in the manuscript (p.21-22). It is not clear why MEMO1 knockout cells are more tolerant to lipid oxidation than the wild type. This question awaits further investigation.

– On lines 400-401, the authors state "Thus, metal coordination in MEMO1 is achieved by H49, H81, and C244, while H192 and D189 do not participate in metal binding in MEMO1". However, in Figure 5A, the ITC curves for H192A and wild-type differ quite a bit-- wouldn't this suggest that H192A has some sort of impact?

The structure of iron bound MEMO1 shows that glutathione forms an electrostatic interaction with H192 (Figure 5C), even though glutathione does not directly participate in iron coordination in MEMO1-Fe complex. Since the ITC experiments were conducted in the presence of glutathione as well, the reduced enthalpy of iron binding in the H192A mutant may reflect lack of glutathione binding to the protein.

Please include statistical analysis for Figure S5-S9.

We believe that the statistical analysis of the cell proliferation rates calculated from this data and shown in Figure 2 is more informative.

[Editors’ note: what follows is the authors’ response to the second round of review.]

Dolgova et al. present a well-written manuscript focused on the mechanism of MEMO1 function in tumor cells. The authors explore whether the mechanism of MEMO1 overexpression in breast cancer, especially TNBC, are related to regulating iron given evidence that MEMO1 binds multiple proteins in the iron regulation pathway. While the data is in part compelling, the claims are based on preliminary evidence using gene linkage and ITC to assert an iron-mediated central role of MEMO1 in tumorogenesis and perhaps metastasis and a mechanism is yet clear. Additional data was provided in the revised manuscript but increases confusion without a substantive contribution to the essential revisions which were at best partially addressed and data was removed from the manuscript, making the current manuscript version incomplete to warrant a version of record.Reviewer #1 (Recommendations for the authors):Dolgova et al. present a well-written manuscript focused on the mechanism of MEMO1 function in tumor cells. The authors explore whether the mechanism of MEMO1 overexpression in breast cancer, especially TNBC, is related to regulating iron given evidence that MEMO1 binds multiple proteins in the iron regulation pathway. While the data is in part compelling, the claims are based on indirect evidence for a central role of MEMO1 in tumorogenesis and perhaps metastasis via its effect on iron homeostasis.The authors have attempted to address the comments raised by the reviewers. However, in some cases, their technical difficulties prevented clarification and in others, there was no additional data added to confirm/refute the idea, leading the authors to remove some of the data presented in the original submission. As a consequence, the manuscript is left weaker rather than stronger. Specifically, the transferrin data has been removed and the conceptual relationship with TFR2 is tempered. The argument that MEMO1 allows cells to accumulate more iron is inconsistent with elevated TFR1 as TFR1 is typically decreased in iron loaded cells. The reason TFR1 would be upregulated as a mechanism for iron accumulation is not made clear. It is also not clear what the meaning is of Figure 4C; while the authors describe superficially that high MEMO cells are more sensitive to ferroptosis, the figure and adjoining text do not make this clear. Taken together, the manuscript is premature and more superficial than previously. While there is some interesting and compelling preliminary data, there is insufficient evidence for a complete story and may be better suited in a cancer journal for a more targeted audience.

We have included the results of shRNA knockdowns of BOLA2 and NCOA4, to address reviewers’ comments about possible link between these proteins and MEMO1 (Figure 2 —figure supplement 5 and lines 235-240).

We would like to point out an important distinction. TFR1 is indeed usually decreased as a response to *iron overload*. However, TFR1 level does increase in response to increased intracellular *iron demand*. This has been observed in cancer cells, osteoclasts, and especially erythroblasts, which accumulate very high levels of iron. The intracellular iron, MEMO1 and TFR1 expression levels in our experiments were all measured at the basal iron concentration in the medium, under the normal cell proliferation conditions. Thus, it is logical to assume that the observed increased level of intracellular iron in the high-MEMO1 cells compared to MEMO1-KO, is not overload, but a response to increased demand. This is consistent with the higher rate of high-MEMO1 cell proliferation (e.g. Figure 1, supplement 5, B) compared to MEMO1-KO. We have clarified that the modest TFR1 upregulation observed in high-MEMO1 cells is consistent with the increased iron demand of these cells (lines 283-286).

This new data strengthens the link between MEMO1 and ferroptosis. Figure 4C shows that the high-MEMO1 cells are much more sensitive to the elevated iron in the medium than MEMO1-KO cells. Ferroptosis inhibitor ferrostatin-1 dramatically increases iron resistance of high-MEMO1 cells, with no significant effect on MEMO1-KO cells. Thus, increased iron toxicity in high-MEMO1 cells is due to MEMO1 role in modulating ferroptosis, which is consistent with higher sensitivity of high-MEMO1 cells to the ferroptosis inducer RSL3 (Figure 4E). This is discussed in the section “MEMO1 promotes ferroptosis via increase in iron concentration in the cell” (lines 339-348). Additional clarification has been added.

We have included statistical significance symbol description in the legend of all figures containing statistical analysis and added statistical analysis results, as requested by reviewer #3, to Figure 1 – supplement 5 and Figure 2 – supplements 1-5.

Reviewer #3 (Recommendations for the authors):The authors did address essential revision #1. They attempted to address essential revision #2 but there were 'technical difficulties' of an unspecified nature. The authors addressed essential revision #3 by tempering their claims of a TFR2-MEMO1 interaction.With regards to my specific recommendations, they addressed most but not all. First, I did ask for asterisks to be defined in the legends, as this is missing from several. They responded that the symbols are defined in the methods, but this makes the paper less accessible to the reader. Adding a one-sentence description to a legend will not compromise the brevity of the manuscript. Second, I asked for statistical analysis to be performed on supplemental figures, and that request was denied.Overall, the authors did address some of the issues raised in the initial review. However, the new data that were added do not help clarify the role of MEMO1 in cellular iron homeostasis.

[Editors’ note: what follows is the authors’ response to the third round of review.]

Overall, the reviewers agree that the has been improved. As you can see below, both reviewers raise the important point of whether MEMO1 is really a direct regulator of iron homeostasis. Based on their suggestions, we would request that the title be changed to better show what is mechanistically known from your studies. Please have a look at the below comments and propose an alternative title that reflects their comments.Reviewer #2 (Recommendations for the authors):Having reviewed this manuscript twice previously, my comments concerning this submission can be brief. In the Abstract, the authors state: "We conclude that MEMO1 is an iron-binding protein that regulates iron homeostasis in cancer cells." The data presented directly and substantially support that conclusion. The data presented are novel in that they are the first to provide any evidence for the function of this protein while definitively providing data on the protein's structure and metal binding properties. Genetically, the protein is linked to a variety of proteins critical to cell iron metabolism, and the protein is linked to "iron behaving badly" in cells – ferroptosis- although the work does not elucidate what that connection is except for the inference that cell content of MEMO1 correlates with the abundance of cell iron and that represents the ferroptotic link. This is inference is plausible albeit not experimentally confirmed by the work presented. Given that the authors are clear that the cellular observations in this work are restricted to "cancer cells" it is inappropriate to criticize the work as not reflecting a possible MEMO1 function in your garden variety enterocyte or hepatocyte. I support publication of this manuscript as is in eLife.Reviewer #3 (Recommendations for the authors):The authors have addressed my comments from the last round of review. However, I view this manuscript only as a first step in understanding the potential role of MEMO1 in iron metabolism in cancer cells, given that the results of this study do not establish the specific function of MEMO1 in iron metabolism. There is the distinct possibility that MEMO1 doesn't play a direct role in iron metabolism, and that the phenotypes uncovered in this study reflect far downstream consequences of altered MEMO1 expression. Along those lines, if the editors do accept this paper for publication, I would make one last suggestion-- revise the title of the paper. Without knowing the specific function of MEMO1, it is too early to declare that it regulates iron homeostasis.

We have changed the title to “MEMO1 binds iron and modulates iron homeostasis in cancer cells”.

In the last sentence of the abstract, the word “regulates” was changed to “modulates” for consistency with the title.